



# Seismic Risk: The Biases of Earthquake Media Coverage

**Maud H. Devès[1,2]\*, Marion Le Texier[3], Hugues Pécout[4] and Claude Grasland[4,5]**

[1] Institut de Physique du Globe de Paris, CNRS UMR 7154, 75238 Paris Cedex 5, France – Université de Paris.

[2] Université Paris-Diderot, Centre de Recherche Psychanalyse Médecine et Société, CNRS EA 3522 – Université de Paris.

[3] Université de Rouen Normandie – UMR CNRS 6266 IDEES, 76781 Mont-Saint-Aignan Cedex, France.

[4] CNRS, FR 2007 Collège international des sciences territoriales – Université de Paris.

[5] Université Paris-Diderot, UMR 8504 Géographie-Cités & FR 2007 CIST, 75006 Paris, France – Université de Paris.

\*Corresponding author: Maud H. Devès (deves@ipgp.fr)

**Abstract**

The capacity of individuals to cope with threatening situations depends directly on their capacity to anticipate what will come next. The media should play a key role in that respect, but an extensive analysis of earthquake media coverage by the international press reveals systematic biases. Exploring a corpus of 320 888 news articles published by 32 worldwide newspapers in 2015 in English, Spanish or French, we found that the press covers a very small number of events: 71% of the news was dedicated to only 3 earthquakes (among the 1559 of magnitude 5+). A combination of frequency and content analysis reveals a typical framing of the 'earthquake news'. Except for the 'Nepal quake', the duration of the coverage is usually very short. The news thus tends to focus on short-term issues: the event magnitude, tsunami alerts, human losses, material damage, and rescue operations. Longer-term issues linked to the recovery, restoration, reconstruction, mitigation and prevention are barely addressed. Preventive safety measures are almost never mentioned. The news on impacts show a peculiar appetency for death counts, material damage estimates and sensationalism. News on the response tends to emphasize the role played by the international community in helping the 'poor and vulnerable'. The scientific content of the coverage is often restricted to mentions of the magnitude, with the concept of the seismic intensity being largely ignored. The notion of the 'seismic crisis' also seems unclear, with aftershocks sometimes being treated as isolated events. Secondary hazards are barely mentioned, except in the case of tsunami alerts. Together, these biases contribute to fatalistic judgments that damage cannot be prevented. If scientific messages are to be communicated, they should be broadcast a few hours after an event. Why not taking that opportunity to familiarize people with the real timeline of seismic disasters?






## Keywords

earthquake, media coverage, seismic risk, risk perception, international news flow theory




## Key Points

- Analysis of earthquake media coverage by the international press reveals systematic biases in the coverage of seismic crises
- News focuses on a small number of events: in 2015, 3 earthquakes attracted 71% of the news (among 1559 earthquakes of magnitude over 5)
- The duration of the coverage is very short with respect to the issues at stake: from a few hours to a few days, rarely more
- The 2015 Nepal quake was exceptionally well covered both in terms of duration and number of news items
- There is a typical framing of 'earthquake news' in the international press
- News content focuses on short-term issues: the event magnitude, tsunami alerts, human losses, material damage, and rescue operations
- Longer-term issues linked to recovery, restoration, reconstruction, mitigation and prevention measures are barely addressed
- To reach the public, scientific messages should be released within hours of big events. Why not taking that opportunity to familiarize people with the real timeline of seismic disasters?










## 1 Introduction

### 1.1 The media play a key role in times of disaster

The coverage that is made of an event has a huge power to influence national and global public opinion, giving (or not) visibility to disaster-related issues. With social media, the online press is among the fastest channels for informing a large number and a great diversity of people. One would expect the press to not only inform but also to empower the communities with relevant knowledge to influence public action and policy toward disaster preparedness and mitigation. Things, however, have proven to be more complex.

Scientists often blame journalists for distorting their messages and for playing the role of a "crisis catalyst" (Boin et al. 2008). Comparing the news treatment of a real earthquake with that of a false quake prediction, Smith (1996) explores the place of science in the media. His study leads him to conclude that "the interest in drama at the expense of public affairs interferes with good scientific reporting." In general, scientists denounce the tendency of the press to search for "culprits" and "accountability" and for "stirring up old rivalry and exaggerating conflicts" (Harris, 2015a and b). Harris (2015a) explores the biases introduced by the 'media filter' in the communication of scientific information during the eruption of Iceland's Eyjafjallajökull volcano in 2010. He shows how the placement of the information in the frame of the pages, selection of stories, use of sources, selection of data, exaggeration, omissions and preferences for certain sources or pieces of information contribute to the oversimplification of complex arguments and an orientation toward information interpretations forcing inclination or prejudice for, or against, an argument, person or group, putting a particular emphasis on some aspects of the situation. Harris (2015b) explores the influence of this media filter on the perception of uncertainty by the public and argues that a careful study of the media filter can help scientists to communicate in a manner that reduces the chance of message distortion.

Numerous studies have explored the ability of the news media to influence public perception. According to McClure et al. (2001) and Mc Clure and Velluppillai (2013), public education programs and news reports often describe disasters "in ways that accentuate the extent and severity of damage", thus contributing to "fatalistic attributions and judgments that the damage cannot be prevented". Improper attribution can hinder peoples' preparedness: "When people attribute damage to an earthquake's magnitude, they invoke an uncontrollable cause, but when they attribute damage to human design, they invoke a relatively controllable cause". For authors such as Gaddy & Tanjong (1987) or Hiroi, Mikami, & Miyata (1985), understanding how the media report on disaster situations has direct implications as it shows "how agencies could reduce fatalism and facilitate preventive action by the way they present information about earthquakes and other disasters."

From the social science and humanities perspectives, media do not just introduce biases into the perception of 'real' events, they also construct part of the reality (Searle & Willis, 1995). Media are primarily seen as being a cultural tool helping people to make sense of what happen to them, collectively. Among the few psychological studies focusing on the impacts of media coverage in a post-disaster context, Yoshida et al. (2016) suggest that watching the





news may even help people to recover from their traumatic experiences, as it provides a good opportunity for deliberate rumination over disaster-related memories. Studying two Canadian rural communities following a forest fire in 2003, Cox et al. (2008) show that the newspaper coverage acts as "a local as well as a broader cultural resource for affected individuals and communities in determining the 'correct' way of responding to and recovering from the disaster". Their analysis emphasizes the power of media "to convey and normalize dominant cultural assumptions" and influence social attitudes and health-related behavior (Gaddy & Tanjong, 1987). It points out the effect of the neoliberal discursive framing of recovery, emphasizing the economical-material aspects of the process and a reliance on experts. Cox and Perry (2011) shows that the dominant discursive constructions of disasters have drawn on and reinforced a hierarchy of credibility in which local voices are marginalized in favor of experts.

### 1.2 This study

This study is led by a pluri-disciplinary team of researchers (from geophysics, psychology and geography). It builds on previous results (Devès, 2015; Grasland et al., 2016; Le Texier & al., 2016) to address the following question: in a globalized world, can we find systematic trends in how the international press covers earthquake events? Many hypotheses about the rules governing the international news flow were formulated more than 50 years ago (Galtung & Ruge, 1965; Östgaard, 1965) and verified by empirical studies concerning the unequal salience of countries in the media and the effects of size, proximity and the preference for elite countries or negative news (Peterson, 1981; Kim & Barnett, 1996; Wu, 2000). The development of new forms of electronic communication has not modified the rules previously observed, and recent works confirmed that the circulation of international news is still very influenced by cultural factors such as language and physical factors such as the distance between the location of the media and the location of events (Segev, 2016; Grasland et al., 2016). However, the salience of countries is generally manifested over a mixture of heterogeneous events, and some authors have focused on subsets of events that are either mentioned or ignored by the media. The event-oriented approach is based on a selection of foreign news related to a specific topic for which it is possible to define a finite and possibly objective list of events occurring in the "real" world. One of the most interesting areas of research from this perspective is the study of the media coverage of earthquakes, for which objective measures of the magnitude or victims are regularly published. It is then possible to analyze the level of newsworthiness according to the different laws postulated by Galtung (Koopmans & Vliegenthart, 2010).

Examining the media coverage of more than 900 earthquakes, Le Texier et al. (2016) showed that the event severity (reported by the media as a moment magnitude) affected the volume of media coverage following a power law. Studying the dynamics of public interest in major earthquakes using Google Trends, Tan & Maharjan (2018) find that the duration and search peak vary with the death toll and damage but not with the earthquake magnitude. Earle et al. (2010) found the same pattern for the 2009 Mw 4.3 Morgan Hill (California) earthquake using Twitter data, in a period of only a dozen minutes.

This study goes further. First, in analyzing the intensity, time distribution and content of a large corpus of approximately 382 249 news items published by 32 international media RSS



feeds in 2015. Second, through the association of a statistical analysis of the news frequency
with a textual analysis of the content of the news. Section 2 presents this dataset and the
methodology we adopted for analyzing it. Section 3 offers a description of our major results,
and Section 4 concludes the paper.

**2 Materials and methods**

**2.1 Presentation of the datasets**
The datasets run from January 1, 2015 at 00:00:01 to December 31, 2015 at 23:59:59.
2015 is particularly interesting as it is the year of the Nepal Quake, a major event well
covered by the international press. The *geophysical dataset* is built from the online seismic
catalogue provided by the USGS (ANSS). The *media dataset* is built from the ANR corpus
GEOMEDIA, which contains information published by more than 330 news RSS feeds from
180 media, localized in 61 countries and written in 10 languages over three years (ANR-12-
CORP-0009, Grasland et al., 2012-2015). We selected international media RSS feeds based
on several criteria: media quality, RSS feed regularity, media localization, and the volume of
transmitted information. The final corpus consists of 32 RSS feeds related to international
news in three languages (English, French and Spanish) that are sufficiently homogeneous and
equitably geographically distributed, according to the possibilities offered by the initial
database (Figure 1).

(insert Figure 1 – currently located at the end of the document)

**2.2 Data cleaning and selection through tagging**
Before starting the data analysis, three processing steps were required (Figure 2). First,
some of the selected RSS items were not worth analyzing because they were totally devoid of
information, simply advertising or summarizing a heterogeneous set of news of the day. These
items were deleted from the corpus. Second, the initial database continuously collects RSS
items on newspaper websites, and a similar item can be published several times without
changes. Therefore, we had to delete all the duplicate items (items with the same title and
text). During these two processing steps, more than 60 000 RSS items were deleted. After the
cleaning, the dataset contains 320 888 news items. To build the joint corpus (called EQ-
MEDIA in the following), we then enriched the media dataset with a tagging process in two
steps: 1) the geographical tagging of all mentioned countries using word dictionaries and 2)
the thematic tagging of all news mentioning a seismic event using an 'earthquake dictionary'.
The first dictionary was tested and validated in previous research (Grasland & al., 2016). The
latter has been tested manually on 1% of the total number of items to determine the number of
false positives (i.e., items containing metaphoric references to earthquakes such as a 'political
earthquake'). We found a reasonable error rate of approximately 4%. The rate of false
negatives (i.e., missed items) was even smaller (approximately 2 to 3%). The final number of
news items dedicated to earthquakes over the year 2015 is 4411, which represents 1.37% of
the total number of items published during that time period by all the RSS feeds of the corpus.
(insert Figure 2)





**2.2 Two levels of analysis: the year 2015 and 3 major events**

An analysis of the intensity and duration of coverage is undertaken on the whole EQMEDIA corpus. The analysis of the news content, which requires coupled qualitative and quantitative approaches, is undertaken on a selection of earthquakes. As shown in Figure 3, the 'earthquake news' is not evenly distributed over time. Three earthquakes garnered the most attention:

- *the Gorkha earthquake:* Nepal and neighboring countries witnessed a 7.8 magnitude earthquake on the 25th of April 2015. It was followed by many aftershocks, among which one on May 12th had a magnitude of 7.3. These earthquakes killed more than 9,000 people and affected at least 8 million, affecting the main economic and political center of the country (Katmandu) and causing massive economic losses (half of the GDP of the country) (CRED, 2017). The first quake (April 25th) was the most devastating. It also triggered landslides and avalanches in the mountains, killing hundreds of people, among whom were foreign tourists whose fates most interested the media. The magnitude of the main shock was similar to that of the 1934 earthquake.

- *the Ilapel earthquake*: An earthquake of magnitude 8.3 hit the area of Ilapel, Chile, on September 9th, 2015, killing at least 15 persons and affecting thousands. Chilean authorities ordered the immediate evacuation of the coast due to a tsunami threat. Pacific-wide tsunami warnings were issued, and the evacuation affected approximately 1 million people.

- *the Hindu Kush earthquake*: An earthquake of magnitude 7.5 hit the Hindu Kush region between Afghanistan and Pakistan on October 26th, 2015. The earthquake and its aftershocks killed approximately 400 people and affected thousands in Afghanistan, Pakistan and the neighboring countries (including India and Tajikistan).

(insert Figure 3)

**2.3 Analyzing the news content**

To more closely examine our dataset, we adopted a method inspired by Cox et al. (2008) who analyzed the print-news media coverage of the recovery process following a forest fire. The first step toward critical discourse analysis is to conduct a careful analysis of the content of the news itself to identify thematic patterns but also possible "textual silences", defined by Huckin (2002) as "the omission of some piece of information that is pertinent to the topic at hand". As we are dealing with hundreds of thousands of items, this qualitative approach is complemented by a quantitative analysis based on keywords.

It was possible but ultimately not relevant to proceed to a classification of the content of our thousands of items with inductive exploratory methods such as cluster analysis (Wilks, 2011) or latent Dirichlet allocation (Blei & al., 2003). Thus, we chose a deductive approach where we tried to extract from the media coverage the categories or concepts defined by experts on disasters. Following Hass, Kates and Bowden (1977) and Kates et al. (2006), we define six *expected categories of content*: hazards, impacts, response, restoration, reconstruction and preparedness. The category of *hazards* refers to the seismic phenomenon itself or to any hazardous event it can trigger such as tsunamis or landslides. The category of



*impacts* refers to the immediate effects of these hazards: human loss, injuries, and damage to
buildings and infrastructures. The category of *emergency response* refers to the actions taken
during or immediately after the earthquake to save lives, reduce health impacts, ensure public
safety and meet the basic subsistence needs of the people affected. The category of
*rehabilitation* includes recovery and restoration, i.e., actions taken to restore basic services
and facilities and improve the livelihoods and health, as well as economic, physical, social,
cultural and environmental assets, systems and activities, of the earthquake-affected
community. By *reconstruction*, we mean the medium- and long-term rebuilding and
restoration of the critical infrastructures, services, housing, facilities and livelihoods.
*Preparedness* refers to actions carried out to build the capacities needed to efficiently manage
future emergencies. News may refer to one or several of these categories of content.

We classify the most frequently used words of the 'earthquake news' into one of these
categories of content and build two keyword dictionaries: a *discourse content dictionary*
corresponding to the above categories (table 1) and an *identity matrix* dedicated to actors
(table 2). For this work to be manageable in a reasonable time, we adopt a threshold of a
minimum of 4 occurrences in French and Spanish and 8 in English (there are, respectively,
619 and 478 items in Spanish and French, so the threshold remains very low, as it corresponds
to words occurring in at least 0.36% of the items. There are 2097 items in English, and thus
the threshold remains sensibly the same: it corresponds to words occurring in at least 0.38%
of the items). Conjunctions and adverbs are not considered, and words with common roots are
treated together. We use words that are representative of one and only one of our categories of
discourse (principle of exclusivity) and that do not introduce too many false positives.
Tagging the database using these two keyword dictionaries allows us to quantify the
presence/absence and evolution of each theme/subtheme/topic. There are limitations to this
keyword approach, but the independent classification of the items by the coauthors indicates a
good consistency in the coding of themes and subthemes and the identification of topics (we
reach a maximum of 12% of differences for the emergency response category).

(insert table 1 and table 2)

**3 Results**

**3.1. 'Earthquake news' analysis of temporality**
News concentrates on a very small number of earthquakes. 71.4% of the items were
dedicated to three earthquakes (Figure 3). The 'Nepal Quake' was exceptionally well-covered,
representing 59.7% of the news, and the earthquakes in Chile (Ilapel) and Afghanistan (Hindu
Kush) collected, respectively, 6.1% and 5.8% of the news. The other events of the year (some
of which are visible as small peaks in the brown curve of Figure 3) share the remaining 28.6%
of the coverage.

The curves of coverage intensity exhibit a similar trend for all earthquakes: the initial
peak is followed by an exponential decrease. This signature has been proved as typical of the





media coverage of dramatic events, characterized by an initial shock to public opinion
(Boomgaarden, H. G. & de Vreese, 2007). The amplitude of the initial peak is higher in the
case of the 'Nepal Quake' than in the other cases. The duration of the coverage is also much
longer with a second peak, corresponding to the aftershock of May 12th, triggering a new
round of coverage. This may be explained by various factors, including a death toll an order
of magnitude higher and that it affected the economic and political center of a touristic
country (Koopmans & Vliegenthart, 2010). However, despite these differences in intensity
and duration, the overall signature of the 'Nepal quake' is similar to the signature of the
Hindu Kush earthquake, likely because both events occurred in similar geodynamical settings
(i.e., intracontinental faulting) and both caused massive impacts (i.e., huge death tolls and vast
material damage). The real question is why the Chilean earthquake, which only caused
moderate impacts, was so well covered. Occurring in a different geodynamical setting (i.e.,
subduction faulting), the earthquake triggered tsunami waves threatening many countries on
the ocean rim. The release of the tsunami alert explains the level of the international coverage
in remote countries. All together, these observations support earlier works showing that the
death toll in itself is not sufficient to predict the volume of media coverage, as other factors –
such as the physical, political, or economic distance to the place of publication – also
influence the newsworthiness of disasters (i.e., Adams (1986), Simon (1997), and Van Bell
(2000), among others).
Eventually, the main peaks of intensity are not significantly different among the English,
Spanish and French newspapers. Only small differences are observed, essentially on the
extent of the main peaks or on the secondary peaks. The similarity of the results obtained in
the three different languages confirms the robustness of our methodology. It also suggests the
existence of a typical *and global* framing of the 'earthquake news', inviting us to dive deeper
into the analysis of content.
**3.2. 'Earthquake news' analysis of content**
**3.2.1 News reproduces the categories of content expected from Disaster Risk**
**Management (DRM) models**
The 'earthquake news' content broadly reproduces the sequence expected from DRM
models but with an important bias: the duration of coverage is too short (hours to days) for
mid- to long-term issues (weeks to months or years) to be well-covered (Figure 4). The
themes of *Hazards, Impacts* and *Emergency Response* are overrepresented compared with
those of *Recovery, Restoration, Reconstruction* and *Preparedness* (Figure 5).
•   77% of the news items contain a general description of the *Impacts* of the event,
either simply to outline its level of destructivity or to count fatalities.
•   46% of the news items refer to the *Hazards*, often to communicate the
magnitude of the earthquake but sometimes to inform about secondary hazards
such as tsunamis, aftershocks and, more rarely, avalanches, mud slides or floods.
•   45% of the news items refer to *Emergency response* describing either aid, search
and rescue operations (in the case of the Nepal and Hindu Kush earthquakes) or
the release and lifting of tsunami warnings (in the case of the Ilapel earthquake).



• Only 5.6% of the news items refer unambiguously to *Recovery, Restoration and*
*Reconstruction*, and none refer directly to issues of *Preparedness*. These low
percentages are partially due to the small numbers of keywords identified for
each of these themes, but it is the low frequency of these themes in the database
that prevented us from identifying more keywords.

It is interesting to note that the big aftershock of May 12th in Nepal triggered a new cycle
of information. Although characterized by a peak of smaller intensity, the news content
followed a similar sequence to the one triggered by the main shock.
Figure 6 show the temporal distributions of these themes. The Nepali and the Afghani
earthquakes have similar signatures: content on hazards comes first, very soon followed by
content on impacts; content on response comes next, and content on recovery, rehabilitation
and reconstruction comes later on – when it comes. The Chilean earthquake has a
significantly different signature, which is due to its tsunamigenic character. The news focuses
first on the hazards including tsunamis, which makes the content on the response (tsunami
warnings) appear much earlier.
(insert Figures 4, 5 and 6)
**3.2.2 The typical 'earthquake news'**
To give a sense of the framing of 'earthquake news', in the following, we build an
(artificial but well-informed) example of the evolution of the news content over time after an
event. Of course, there are to be variations due to elements of context, but our guess is that the
main trends would remain comparable.
Imagine that an important earthquake occurs…
• **Within a few hours**
The news focuses on the description of the seismic hazard and, when relevant, passes on
information about tsunami warnings. The news first reports that an earthquake has been felt,
providing an approximate location of the impacted area (often a country, sometimes a region
or a city). Many recall the magnitude of the event.
e.g., 'USGS: Magnitude 7.5 earthquake strikes Afghanistan' (USA today, October 26th, item
10366718), 'Un terremoto de 7,9 grados sacude el centro de Nepal' (Faro de Vigo, April 25th,
item 6369528), 'Un séisme de magnitude 7,5 a secoué lundi le massif de l'Hindu Kush' (Le
Monde, October 26th, item 10368842)

It quickly becomes clear that the event is worth mentioning because it had noticeable impacts.
e.g., 'La ONU advierte dramático impacto tras nuevo temblor en Nepal' (El informador, May
13th, item 6774985), 'Scores of people were killed when a 7.5-magnitude earthquake centered
in Afghanistan rocked neighboring Pakistan and rattled buildings as far away as India.' (USA
372         Today, October 26th, item 10371195)

The combination of the location and magnitude is often use to 'label' the event and
distinguish it from other ones. After a few days, 'big' events are known by their 'nicknames',



and the magnitude is less often mentioned. A few hours after the main shock, journalists
named the earthquake the 'Nepal earthquake', and it soon became the 'Nepal Quake'.
e.g., '5 things to know about the Nepal earthquake' (The Star, April 25th, item 6376436) 'Nepal
quake: 7.9 magnitude tremor hits near Kathmandu' (The Guardian, April 25th, item 6370804)
However, only a few earthquakes become famous enough to be called by nicknames; the
Chilean and Afghani earthquakes of 2015 did not, and the news settled for recalling the
country and magnitude of the main shocks.

Interestingly, that initial phase of coverage is also the phase with most scientific content.
The extensive use of the notion of magnitude, although often made at the expense of the
notion of seismic intensity, testifies to the successful transfer of a geophysical notion to the
lay public. We should also outline here that aftershocks are sometimes treated as singular
events by the press, with the notion of a seismic crisis remaining unclear to many. Among the
most cited expert bodies, the USGS is the most visible internationally, as it provides
immediate information about the earthquakes. Regionally important centers such as the
Servicio Hidrográfico y Oceanográfico de la Armada (SHOA) in Chile can also be cited.

Secondary hazards are barely mentioned in the news, except for tsunamis. In Chile, the
news passed on very well the information about tsunami warnings, mentioning at the same
time the primary and the secondary hazards and the authorities' response to it:
e.g. 'Tsunami warnings in Chile and Peru as 8.3 quake hits' (Daily Telegraph, September 17th,
item 9501990), 'The tsunami warning from New Zealand's Ministry of Civil Defence & Emergency
Management after a big quake off Chile will affect a night surfing event.' (The Age, September
17th, item 9504366).

•   **Few hours to few days after the event**
The peak of coverage is reached within a few hours to a day after the event, with many
updates of the same news including more and more precision or detail. Earthquake events
become 'breaking news' or 'top stories' and are disseminated simultaneously on different
RSS feeds. Most news talk about impacts, especially human losses. The description of the
impacts is the theme that attracts the most coverage. 76.7% of the news of our corpus focuses
on the description of the impacts (81% for the three considered earthquakes). 34.3% focus on
human losses, and only 17.3% on material damage. Messages about human impacts adopt a
factual tone and evolve following a rather systematic pattern.
For illustration, we provide an example of the treatment by *The Guardian* of the 'Nepal
Quake'. The news starts by mentioning the occurrence of an event with fatalities:
e.g., 'Fatalities as earthquake hits Nepal' (The Daily Telegraph, April 25th, 09:19, item 6371294)
Within a few hours, the regular update of the human losses starts:
e.g., 'Nepal earthquake: more than a hundred people dead' (The Guardian, April 25th, 12:04,
item 6371816), 'Nepal earthquake: nearly 700 people dead' (The Guardian, April 25th, 13:42,
item 6373501), 'Nepal quake: more than 1,000 people dead after tremor near Kathmandu'
(The Guardian, April 25th, 17:44, item 6381853)
As the hours go by and the numbers continue to rise, concurrent topics start emerging.
Stories become more personalized, the event starts to be romanticized and the news starts
referring to distinct categories of victims (famous people, nationals, vulnerable ones, etc.):



e.g., 'Nepal quake kills more than 1,000 and spreads terror on Everest' (The Guardian, April 26th, 00:23, item 6382569), 'Google executive Dan Fredinburg filmed at Everest base camp before death' (The Guardian, April 26th, 16:49, item 6396313)), 'Népal: le bilan des victimes françaises pourrait s'alourdir' (Le Parisien, May 3rd, item 6542461)

Aid and rescue operations and life conditions start attracting interest:

e.g., 'Nepal earthquake: rescue continues as death toll exceeds 2,500' (The Guardian, April 26th, 18:18, item 6397229), 'Nepal earthquake: thousands seek shelter as death toll exceeds 2,500' (The Guardian, April 27th, 2:04, item 6402976)

As the days go by, the death toll appears less frequently, with the news reporting official numbers only when those are updated:

e.g., 'Nepal earthquake death toll exceeds 4,000 with many still missing. More than 4,000 are confirmed dead and 6,500 injured…' (The Guardian, April 28th, item 6430398)

Proportionally, there is a lack of interest in injuries and general health issues (with psychological issues even more ignored).

During the phase of coverage dedicated to impacts, we observe a tendency to sensationalism. Almost half of the news items use superlatives such as 'devastating', 'powerful', 'catastrophic', 'enormous', 'dramatic', 'monster', or 'violent', etc., emphasizing the extent of the devastation. Surprisingly, terms referring directly to emotions (such as 'fear', 'desperation', 'panic', 'courage', etc.) remain rare.

e.g., 'Nepal's second monster quake' (The Australian, May 12th, item 6749166), 'As rescue efforts were hampered by bad weather, dramatic details emerged about the devastation at the base camp in the wake of an avalanche' (The New York Times, April 28th, item 6423784), 'Nepalíes cavaron con sus manos para sacar a sobrevivientes de montañas de escombros. Pánico. Lágrimas. Miedo. Todos estos sentimientos se conjugaron ayer como parte de la jornada trágica que vivieron los miles de nepalíes que habitan Katmandú, y es que tras el fuerte terremoto de 7.8 grados en la escala de Richter que dejó en el país al menos mil 475 muertos […] los sitios históricos están completamente devastados' (La chronica de hoy, April 26th, item 6387254), 'vías de comunicación completamente sepultadas por corrimientos de tierra y rocas' (La chronica de hoy, October 27th, item 10394058), 'En el barrio de Gongabu, completamente arrasado, fallecieron 500 de las 8.000 víctimas del terremoto' (El Pais, May 13th, item 6779435), 'Reportage dans des villages coupés du monde, dévastés par la catastrophe, où les secours peinent à arriver comme l'aide des autorités.' (Le Monde, April 28th, item 6434796)

- Within a few days after the event

The focus slides from impacts to response operations. 45.2% of the news of our corpus refer to that category (Figure 5). In the case of a tsunami alert, the theme of response operations appears earlier in the coverage, as the news passes on information about warnings and, if relevant, mass evacuations. In the absence of a tsunami threat, the news focuses on aid, search and rescue operations. In that case, evacuation and displacement are generally undercovered.

e.g. 'Rescue teams dig for Nepal quake survivors' (USA Today, April 27th, 6401498); 'Rescuers were struggling to reach quake-stricken regions in Pakistan and Afghanistan on Tuesday as officials said the combined death toll from the previous day's earthquake rose to 339.' (The Times of India, October 27th, item 10393016), 'FRANTIC rescue efforts to save people trapped under rubble are taking place after a 7.9 magnitude earthquake hit near Nepal's capital, Kathmandu.' (Daily Telegraph, April 25th, item 6372184)

First, the messages adopt a general tone, becoming more specific when the international community starts sending help:



e.g., 'China's rescue team pulls first survivor out of debris after Nepal quake ' (China Daily, April
27th, item 6409965), 'The burly Californian and fellow members of a disaster response team
deployed by the U.S. Agency for International Development were looking, against all odds, for
collapsed buildings' (The Los Angeles Time, May 1st, item 6499637), 'Turkish rescue workers in
Kathmandu, Nepal pulled a man alive from the rubble of a destroyed building on Monday.' (USA
Today, April 27th, item 6414192).

We note a tendency of the international press to glorify the contribution of the
international community to help the 'poor and vulnerable'.

Rescue operations are also an occasion for relating personal stories, if not miraculous
ones.
e.g., 'Google executive Dan Fredinburg filmed at Everest base camp before death' (The Guardian,
April 26th, item 6396313), 'Boy found alive 5 days after Nepal quake' (The Age, April 30th, item
483 6481498)
Such stories, that one could call *topoi*, can take different forms depending on context. In
Nepal, one finds several stories about 'children saved from the rubble' (The Guardian, April
30th, item 6480552). In Afghanistan, stories focus on 'twelve girls caught in a stampede while
trying to escape from their school' (Daily Telegraph, October 26th, item 10367166).

At that stage, the duration of coverage plays an important role in the richness of the
content of the news. The coverage of the 'Nepal Quake' is longer and richer: the living
conditions, internal displacement, epidemic risk, and mass cremation are all issues that are not
at all addressed in the coverage of the other earthquakes.

• **Few days to few months after the event**
The coverage intensity has faded out, impeding the proper coverage of long-term issues
(Figure 4). Few items refer to *recovery*, which tends to cover distinct temporalities, from a
few days to several months (Figure 5).
e.g., 'Nepalese villagers clean up four days after a monster earthquake killed more than 5,000
people in the Himalayan nation' (USA today, April 29th, item 6462063), 'The International
Federation of Red Cross and Red Crescent Societies warned on Friday that longer-term support is
needed to help shattered communities recover six months after a magnitude 7.8 earthquake
struck Nepal.' (China Daily, October 10th, item 10361489)
The theme of *reconstruction* is dedicated to more permanent repairs and rebuilding.
There are enough items referring to that theme for us to identify a few keywords, but the
coverage remains poor (Figure 5). There are again different temporalities. In the short term,
the news reports that people are rebuilding their homes. In the longer term, the news reports
the reopening of public infrastructures such as schools, hospitals and historical buildings as a
sign of returning to normal life.
e.g., 'Survivors in quake-hit Pakistan seek help to rebuild homes' (Times of Malta, October 28th,
item 10408082), 'Hundreds of thousands of Nepalese children have returned to school in Nepal
for the first time since two earthquakes last month killed more than 8,700 people and injured
23,000…' (The Guardian, May 31st, item 7161853)




- A window of communication for scientists

According to Haas et al. (1977), the second and longer phase of reconstruction corresponds to the continuing assessment of hazards and risks and structural and nonstructural improvements to reduce the impact of future events (i.e., mitigation and adaptation measures, prevention). This phase lasts many years, during which attempts are made not only to recover but to improve the state of living, and society devotes some attention to the construction of memorials or the institutionalization of a narrative memory of the event. We could not find enough items referring to mitigation, adaptation and prevention to identify keywords. There are, however, a few items referring to a narrative dimension: the ones that place the event in a country's history.

> e.g., 'El terremoto fue el sexto mayor movimiento telúrico en la *historia* de Chile y el de mayor
> intensidad en el mundo durante 2015.' (El Universal, September 17th, item 9516610)

A few items also mention the lessons learned (or not learned) from past events.

> e.g., 'Nepal earthquake: learn lessons or more will die in future disasters, warns expert' (The
> Guardian, April 29th, item 6460947), 'How Nepal can avoid the mistakes of Haiti' (The Guardian,
> May 12th, item 6745299)

By doing so, the press contributes to maintaining a form of knowledge about existing risks. That contribution to the collective memory often happen just after the main shock (or after large aftershocks). It is also a time when the press listens to experts, and so it might be a good window for communication. People are looking for elements to make sense of what has just been going on. Scientists can take that chance to send a message.

### 3.2.3 The figures of 'earthquake news'

The identity matrix allows the identification of the categories of actors that are the most present in the news. 44.2% of the news mentions the people affected by the earthquake. The exact terminology varies with time. 'Those affected' start as 'victims' to become 'rescued', 'survivors' and then 'locals' or 'villagers'. 6% of the news refers explicitly to vulnerable persons.

27.7% of the news mentions state representatives who are responsible for organizing the public response, but regional and local public services are absent (Figure 5). Surprisingly, only 8% of the news refers to civil and military security services and 7.7% to rescuers in general. 3.8% of the news mentions UN agencies, and 2.5% international aid. Only 5.4% of the news refers to experts, specialists or scientists, mostly during the initial phase of coverage after the main shock and after the big aftershock in the case of the Nepal Quake. The private sector is rarely mentioned, except Google and Facebook for their people finder tools. Other figures emerging from the 'earthquake news' are 'famous unknowns' whose stories serve to exemplify the experience of the affected people. The news sometimes refers to famous personalities, either because they are among the victims or because of their generous donations. It is interesting to observe that local communities and their representatives are almost absent from the news.



**4 Discussion**

Studying earthquake coverage at the global scale, we reach different conclusions from authors such as Rovai and Christine (1998). Among the 7 136 earthquakes of magnitude 4.5+ occurring in 2015, we indeed observe significant differences in coverage: most events are not reported by the media, except a few that are particularly well-covered. However, once events are covered, we observe an astonishing homogeneity in the news content. There are, of course, variations in the way journalists treat the information - editorial choices and cultural proximity with the impacted countries are both parameters influencing the duration and content of the coverage - but these variations remain small. Our results suggest that there is a typical framing of earthquake news in the international press.

This framing seems to introduce major biases in the representation of the seismic risk. A first bias is linked to the short duration of the coverage. Analyzing Googling trends, Tan et al. (2018) confirm our empirical observation that the peak of public interest after destructive earthquakes follows an exponential temporal decay. The same tendency was observed for smaller events by Earle et al., 2010. Our results complement these findings in showing that the international online journals follow the same tendency. However, we go further than previous studies in exploring the consequences of that exponential decay on the news content. It focuses the information on short-term issues such as the description of the hazard and of its impacts and emergency operations. The mid-term and long-term issues of recovery, restoration, reconstruction, adaptation, mitigation and preparedness are largely undercovered.

This finding outlines the necessity for scientists to communicate, whenever possible, within a few hours after the occurrence of an earthquake, especially the big ones that are the most capable of catching a large audience. Of course, the need for reactive communication should not result in unpreparedness. Having a knowledge of the content and the evolution of typical earthquake news can help design typical communication tools that could be quickly adapted on a case by case basis once the event has occurred. Designing scientific messages, one should pay particular attention to counterbalance the known biases.

Communicating about the hazards, for instance, it would be important not to insist on including information about the magnitude but to find simple words to pass on the notions of seismic intensity, seismic crisis and potentially earthquake swarm. About impacts, our analysis supports the statement of McClure et al. (2001): the representation of the seismic risk that is built by the press emphasizes the immediateness and hyperdestructivity of the event, occulting the real timing of such disasters: a time to anticipate and get prepared, a time to protect and a time to recover and reconstruct. We agree with Lamontagne et al. (2016): scientific messages should encourage people to take preparedness actions and get them prepared for potential losses, describe to them the timeline of the disaster cycle and teach them ways to diminish losses.

Although unprecedented, we are aware that our study also has some caveats. The use of keywords to quantify themes and topics provides robust conclusions but is not completely satisfactory. We tried to get around its limitations by preselecting words from a list of the most frequently used terms. A further step is to engage with more complete techniques of text analysis combining inductive and deductive approaches. We could, for example, use machine learning methods such as word2vec (Le & Mikolov, 2014) for the simplification of the collection of keywords and the quantification of the different steps of the news coverage.





However, this tool would complement but not replace the qualitative analysis of the content we undertook in this study.

One of our working hypotheses was to demonstrate the existence of a global framing of earthquake news and, to reach that goal, we chose to work on the international press, but it would be important to undertake a similar analysis on the national and regional press as well as social media. A recent work by Jamieson and Van Belle (2019) suggests for instance that the level of development of the disaster-stricken community influences the nature of news coverage in other at-risk communities : "if an earthquake occurs in a community with a high level of development, the news coverage is much more likely to draw lessons for their community, and less likely to emphasize differences that prevent policy learning".

Another interesting lead to explore would be to study the evolution of the public state of mind as they read the news. This could allow choosing more carefully which information to provide and at which time (see Wein et al., 2016, for an example).

**5 Conclusion**

"Most people do not experience disasters first-hand, but rely on mediated depictions of distant events." (Jamieson and Van Belle, 2019). This is why it is of utmost importance to study the narratives built by the news media in reporting about distant disasters. In this paper, we explore the media coverage of seismic events in the international press during the year 2015, analyzing 320 888 news published in English, Spanish or French by 32 RSS feeds distributed worldwide. Among the 7 136 earthquakes of magnitude 4.5+ occurring that year, three were predominantly covered: the sadly famous 'Nepal quake' that hit the valley of Kathmandu in April, an earthquake in Chile that shook the area of Ilapel in September, and an earthquake in Afghanistan that struck the Hindu Kush in October. We compare the duration and content of the media coverage of these three major earthquakes with classical models of Disaster Risk Management.

Doing so, we demonstrate that: 1) there is a typical framing of the news about earthquakes in the international press, 2) this framing introduces major biases in representation, impeding the proper appropriation of the seismic risk by the public. The news content faithfully follows the succession of phases predicted by the DRM scheme, describing the hazard before reporting on its effects and the response of the impacted communities. However, an important bias is introduced by the very short duration of coverage: only the first phases of the DRM scheme are covered, while the issues of recovery, restoration, reconstruction, adaptation, mitigation and preparedness remain largely ignored. We also observed the following biases: i) The news tends to concentrate on the description of impacts and, among them, more specifically on human losses. That focus is associated with the pervasive use of sensationalistic terms describing a landscape of devastation, which may contribute to fatalistic judgments that the damage cannot be prevented. ii) The second theme of interest – the second in terms of coverage intensity but the first one in terms of timing - is that of hazards. The communication is centered on the notion of magnitude, with the concept of seismic intensity being ignored. Aftershocks can be occasionally treated as isolated events, testifying to a lack of understanding of the concept of the seismic crisis and, except for tsunamis, secondary hazards are barely mentioned. iii) The third theme of interest is that of



the emergency response. The focus is made on alert and evacuations in case of tsunami warnings and on aid, search and rescue otherwise. Other issues such as safety measures, temporary housing, water or electricity cuts, etc., and longer-term issues are barely mentioned.

On the basis of that analysis, we discussed leads to improve the scientific communication on earthquakes. Taking the opportunity of the short window of interest that follows big earthquakes, scientists should familiarize people with the real timeline of a seismic disaster cycle… which tends to last longer than the interest of the news media.

## Data and Resources

This paper has benefited from the database GEOMEDIA produced and maintained by the International College of Territorial Science (http://www.gis-cist.fr). Earthquake parameters were obtained from the USGS Comprehensive Earthquake Catalog (ComCat), which was searched using https://earthquake.usgs.gov/earthquakes/search/ (last accessed on November, 1[th] 2019).

## Authors contribution

Conceptualization, project administration, methodology, writing – original draft: M. Devès; Writing – review & editing: all authors; Data curation and investigation: M. Devès, M. Le Texier. H. Pécout; Formal analysis: M. Le Texier, H. Pécout; Validation; M. Le Texier, M. Devès; Visualization: H. Pécout, M. Devès ; Resources: C. Grasland.

## Acknowledgements

This paper is a contribution to the Cross-disciplinary Program *Politics of the Earth* of the Université Sorbonne Paris Cité (Université de Paris), to CRPMS and IPGP (contribution number 4031).





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




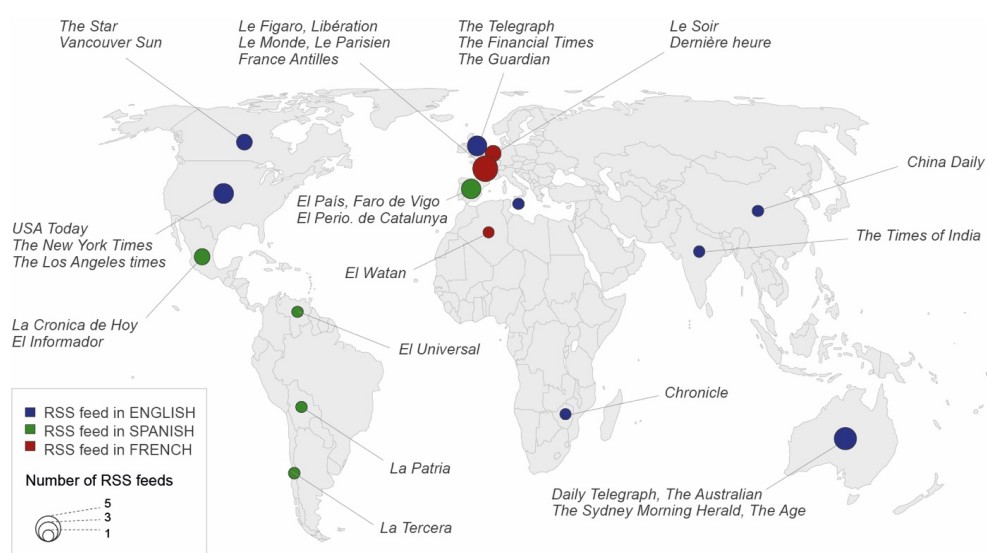


**Figure 1**. Corpus of news RSS feeds used, by origin and language




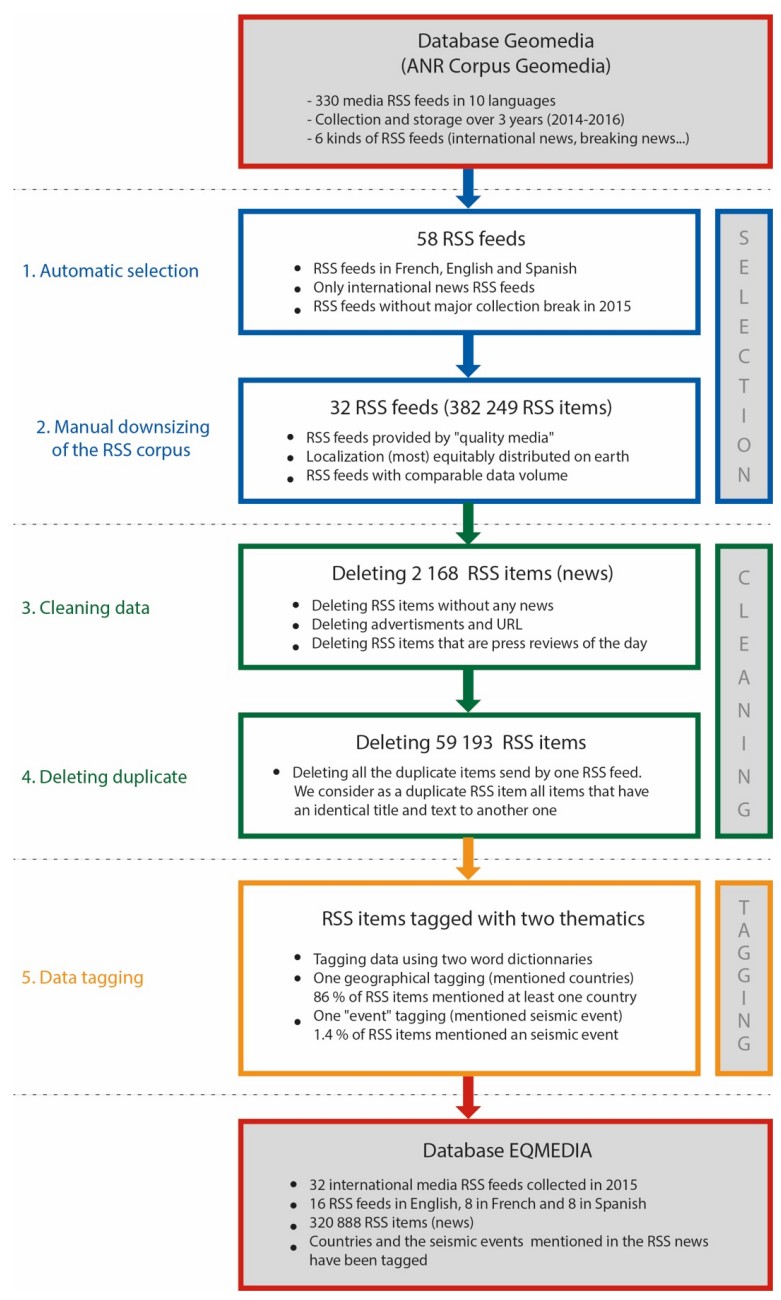

**Figure 2.** Building the EQMEDIA database





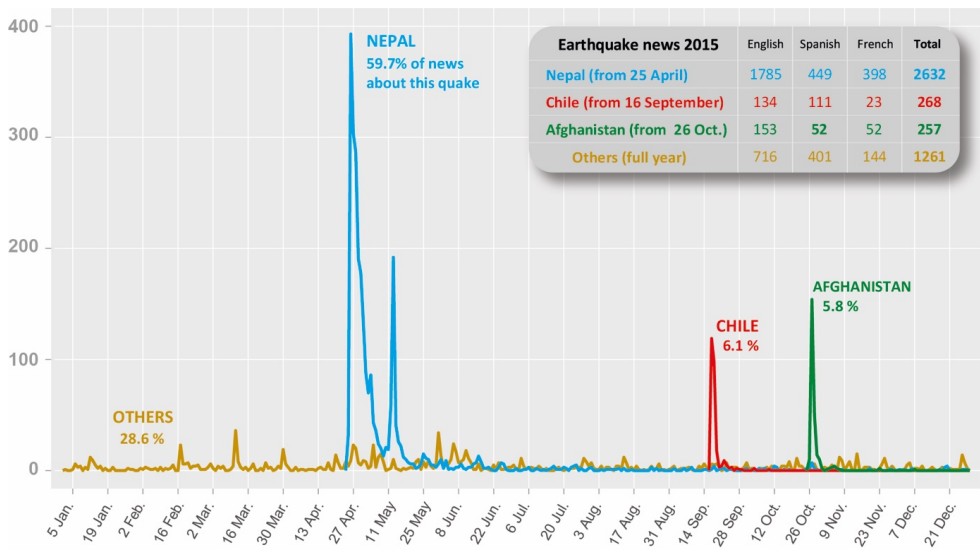

**Figure 3**. The media coverage intensity (number of news articles published per day) of the year 2015 is dominated by three events: the Nepal Quake, an earthquake in the area of Ilapel, Chile and an earthquake in the Hindu Kush, Afghanistan.

**Table 1 (next page).** Discourse content dictionary. Contains the keywords used to classify items into categories of discourse corresponding to the main phases and topics of disaster risk management. Keywords were identified from a list of most frequent words using different thresholds for English, Spanish and French to balance differences in the RSS feed numbers.




| CONTENT CATEGORIES | KEYWORDS BY THEMES AND TOPICS |
|---|---|
| **HAZARDS** | **Magnitude**<br>EN: magnitude, Richter<br>SP: grados, Richter, magnitud(es)<br>FR: magnitude, Richter<br><br>**Tsunami**<br>EN: tsunami(s)<br>SP: tsunami(s), maremoto(s), olas<br>FR: tsunami(s)<br><br>**Aftershocks**<br>EN: aftershock(s)<br>SP: aftershock(s), réplica(s)<br>FR: aftershock(s), réplique(s)<br><br>**Other secondary hazards**<br>EN: avalanche(s), landslide(s), flood(s)/flooding<br>SP: avalancha(s), deslizamiento(s), alud, inundacion(es)<br>FR: glissement(s) de terrain, avalanche(s) |
| **IMPACTS** | **Impacts – general**<br>EN: hit(s), struck, felt, shook, shak(e)(ing)(en), rocked, jolt(s)(ed), rattled, shattered, sway(ed), battered, suffered, toppling, crushed, strike, stricken, impact<br>SP: impacto, estimacion(es), afectación, sacud(e)(ido)(ida)(idas)(ieron), golp(e)(eó)(ea), golpead(o)(os)(a)(as), azotó, azotado, sentido, se sintió, afectó, sufrieron, arrasó, temblar, asoló, castigad(o)(a)<br>FR: frappé(e), touché(s), ressenti(e), ébranlé, secoué<br><br>**Human impact**<br>*Human impact – general*<br>EN: fatalities, casualt(y)(ies), victim(s), affected, stranded<br>SP: balance, víctima(s), afectados, damnificados, recuento(s), saldo, contabilizado<br>FR: bilan, victime(s), sinistrés<br>*Human impact – death toll*<br>EN: death(s), kill(s)(ed)(ing), dead, bodies, died, deadly, claimed<br>SP: muerto(s), muerte(s), mueren, murieron, mortal(es), fallecido(s), fallecieron, cuerpos, cadavers, decesos, mató<br>FR: mort(s), tué(e)(s), corps, meurtrier<br>*Human impact – injured*<br>EN: injured, wounded<br>SP: heridos<br>FR: blesses<br><br>**Material damage**<br>*Material damage – general*<br>EN: rubble, damage(d), collaps(e)(es)(ed) (ing), devastat(ed)(ion), destroy(ed)(ing), destruction, wreckage, debris, ravaged, ruins/ruined<br>SP: daños, escombros, dañad(os)(as), destruid(o)(os)(as), perdidas, destrucción, ruinas, caíd(o)(a), destruyó, destrozadas, colapso, devastó, devastadas, derrumb(e)(es)(aron)(ado)<br>FR: dévast(é)(ée), décombres, dégâts, détruit/détruits, effondr(ée)(ées), destructions, gravats |



| | |
|---|---|
| | *Material damage - on buildings*<br>EN: homes, building(s), houses, structure(s), property<br>SP: edificio(s), vivienda(s), edificaciones<br>FR: maisons, bâtiments<br><br>*Material damage - on infrastructures*<br>EN, FR: no recurrent keywords were found<br>SP: eléctricas, infraestructuras |
| **EMERGENCY RESPONSE** | **Tsunami warning**<br>EN: tsunami warning(s), alert(s)<br>SP: alerta de tsunami, alarma<br>FR: alerte<br><br>**Evacuation**<br>EN: evacuat(e)(ed)(ion)(ions), evacuees<br>SP: evacuad(os)(as), evacuar, evacuación<br>FR: evacu(ees)(er)(ation)<br><br>**Aid, Search & Rescue**<br>*General*<br>EN: effort(s), response, respond, operation(s), deployed, aid, rescu(e)(es)(ed)(ing), relief, help(ed)(ing), assist(ance), helicopter(s), chopper, aircraft, support, send(s)(ing), save(d), distribut(ing)(ion), airlifted, dig(ging), dug, missing, search(ing), alive, pulled, trapped, recovered + table 2/rescuers<br>SP: operación/operaciones, gestión, respuesta, solidaridad, crisis, apoy(o)(ar), ordenó, responder, envoi, enviado(s), reacción, ayuda, ayudar, ayudas, ayudando, rescate, rescatar, rescatan, rescatado, helicóptero(s), asistencia, socorro, attender, ofrece, aeronave, búsqued(a)(as) + table 2/rescuers<br>FR: operation(s), répondre, secours, aide, sauver, assistance, disparu, chiens, recherchés, sans nouvelles + table 2/rescuers<br>*Vital needs and supplies*<br>EN: food, hungry, sanitation, water, drink(ing), fuel, blankets, gasoline, suppl(y)(ies), resources, basic, vital, lack of, goods, need, needed, material, equipment<br>SP: agua, alimentos, alimentaria, necesidad(es), comida, suministro(s)<br>FR: de materiel, besoins<br><br>**Medical care**<br>EN: hospital(s), medical, medicine(s), disease(s), health, outbreak, epidemic(s), treatment, patients<br>SP: hospital(es), médico(s), salud, medicinas, sanitarios<br>FR: no recurrent keywords were found<br><br>**Displacement & Temporary shelter**<br>EN: shelter(s), outdoors, sleep, sleeping, homeless, refuge, fled<br>SP: noche al raso, albergues, tiendas de campaña, desplazados, refugio(s)<br>FR: camps, fuir, dehors<br><br>**Cremation**<br>EN, FR: no recurrent keywords were found<br>SP: funerarias |
| **RECOVERY REHABILITATION** | **Recovery/Reconstruction**<br>EN: recover(y)(ing), return to, returned, reconstruction, rebuild(ing), reopen(s)(ed), |



| RECONSTRUCTION (PREPAREDNESS) | normal<br>SP: desescombro, reconstrucción, reconstruir, normalidad<br>FR: reconstruction<br><br>*No recurrent keywords were found that unambiguously refer to Risk assessment, development and land use planning / Adaptation and mitigation measures / Education and information / Preparedness, contingency planning, consolidate preparations for next disasters* |
| --- | --- |

**Table 2.** Identity matrix. Contains the keywords used to quantify the presence/absence of
different categories of stakeholders. Keywords were identified from a list of most frequent
words using different thresholds for English, Spanish and French to balance differences in the
RSS feed numbers.

| CONTENT CATEGORIES | KEYWORDS BY THEMES AND TOPICS |
| --- | --- |
| **STATES** | EN: nation, state(s), government(s), authorities, minister(s), ministry, foreign secretary, foreign office, president, parliament, royal rulers, embassy, European Union<br>SP: país, nación, gobierno, autoridades, ministerio, ministro, president(a)(e), exteriores, funcionarios, gabinete, ispr, fata, europea<br>FR: pays, gouvernement, affaires etrangeres, autorités, ministère, ministre, Quai d'orsay |
| **UN AGENCIES** | EN: United Nations, UNICEF, UNESCO, World Food Programme<br>SP: onu, naciones unidas, Programa Mundial de Alimentos, unesco, unicef<br>FR: nations unies, onu |
| **INTERNATIONAL AID** | EN: international aid, international agencies, aid agencies, humanitarian aid<br>SP: ayuda internacional, comunidad internacional, organización no gubernamental, ong, cruz roja<br>FR: aide internationale, croix rouge, humanitaire(s) |
| **CIVIL SECURITY & DEFENSE** | EN: police, army, military, marine(s), air force, soldiers, troops, firefighters, Gurkhas<br>SP: ejército, policía, militares, armada, marina, soldados, Oficina Nacional de Emergencia |
| **RESCUERS** | EN: rescuers, rescue team(s), aid workers, rescue workers, relief workers, volunteer(s), personnel<br>SP: equipo de rescate, equipos de rescate, servicios de emergencia, rescatistas, socorristas<br>FR: équipe, secouristes, sauveteurs |
| **AFFECTED PEOPLE** | **Directly affected ones**<br>EN: people, rescued, survivor(s), victims, those affected<br>SP: persona(s), víctima(s), los afectados, damnificados, desaparecid(o)(a)(os)(as), supervivientes, sobrevivient(e)(es), rescatad(o)(os)<br>FR: victimes, survivant(s), sinistrés, rescapes, personnes |



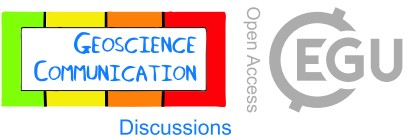

| | |
|---|---|
| | **Locals**<br>EN: residents, locals, villagers, sherpa(s), guides, Famous locals: Ang Tshering, Bajracharya<br>SP: población, habitantes, guías<br>FR: habitants, villageois, population<br>**Vulnerable ones**<br>EN: children, child, boy, girl(s), wo(man)(men), famil(y)(ies), teenag(e)(er), teen, bab(y)(ies)<br>SP: niños, famili(a)(as), muj(er)(eres), jóven, bebe, anciano<br>FR: familles, adolescent, enfants, orphelins |
| **'EXPERTS'** | EN: expert(s), US Geological Survey, specialists, scientists<br>SP: usgs, Centro Sismológico Nacional, especialistas, Servicio Hidrográfico y Oceanográfico de la Armada<br>FR: usgs, institute américain de géophysique |
| **PRIVATE COMPANIES** | EN: Google, Facebook, compan(y)(ies)<br>SP: google, Facebook<br>FR: no recurrent keywords were found |








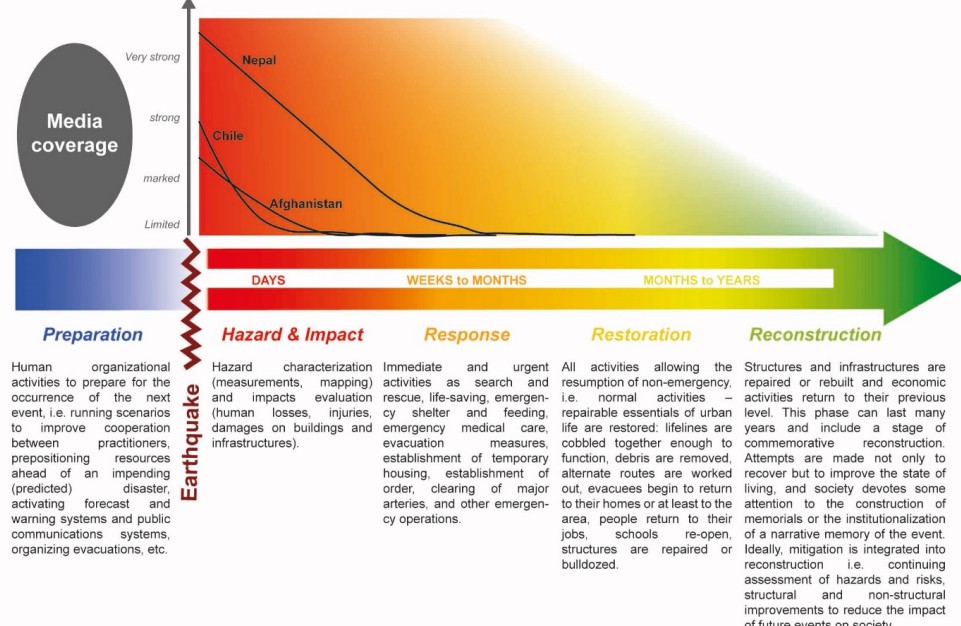

**Figure 4.** Temporal distribution of the media coverage for the three well-covered earthquakes
of the year 2015. The color scale allows comparing the duration of the media coverage with
the expected duration of the different phases of disaster risk management models.




| | Themes | % of earthquake news | Number of items | Subthemes | % | Number of items | Topics | % | Number of items |
|---|---|---|---|---|---|---|---|---|---|
| **Discourse content** | **Hazard** | **45,8** | 2020 | **Tsunami** | **8,9** | 391 | | | |
| | | | | **Aftershocks** | **5,8** | 254 | | | |
| | | | | **Secondary hazards** | **7,8** | 343 | | | |
| | | | | **Magnitude estimation** | **23,5** | 1036 | | | |
| | **Impacts** | **76,7** | 3384 | **General impact** | **40,9** | 1802 | | | |
| | | | | **Human impact** | **49,6** | 2189 | General | 17,1 | 756 |
| | | | | | | | Death toll | 40,7 | 1797 |
| | | | | | | | Injured | 8,9 | 393 |
| | | | | **Material damage** | **30,8** | 1358 | General | 26,1 | 1150 |
| | | | | | | | Buildings | 13,3 | 585 |
| | **Response** | **45,3** | 1996 | | **4,3** | 191 | | | |
| | | | | **Evacuation** | **2,1** | 93 | | | |
| | | | | **Aid Search Rescue** | **34,0** | 1501 | General | 29,6 | 1306 |
| | | | | | | | vital needs | 4,4 | 196 |
| | | | | **Medical care** | **2,2** | 95 | | | |
| | | | | **Temporary shelter** | **2,7** | 117 | | | |
| | **Reconstruction** | **5,6** | 249 | | | | | | |
| **Identity Matrix** | **States** | **27,7** | 1220 | | | | | | |
| | **Un agencies** | **3,8** | 168 | | | | | | |
| | **International Aid** | **2,5** | 111 | | | | | | |
| | **Civil Security Defence** | **8,0** | 353 | | | | | | |
| | **Rescuers** | **7,7** | 341 | | | | | | |
| | **Affected People** | **44,2** | 1951 | Directly affected ones | **33,4** | 1475 | | | |
| | | | | Locals | **4,8** | 211 | | | |
| | | | | Vulnerables | **6,0** | 265 | | | |
| | **Expert** | **5,4** | 239 | | | | | | |
| | **Private Companies** | **1,6** | 72 | | | | | | |

**Corpus = 320 888 news, including 4 411 news about earthquake (1,37%)**


**Figure 5.** Percentage of news by themes and topics





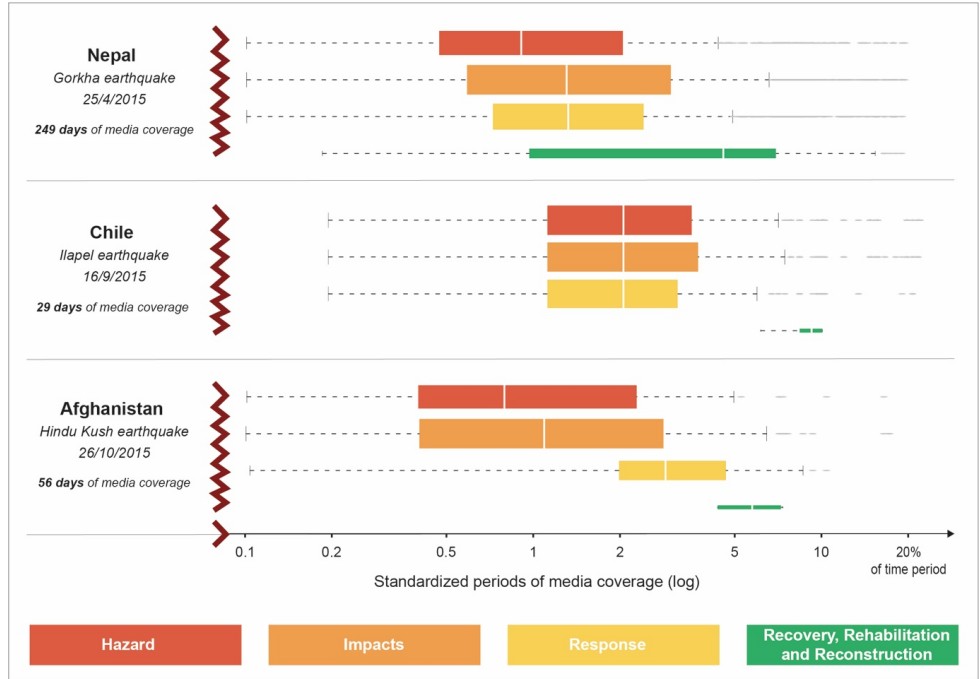

**Figure 6.** Temporal distribution of the DRM categories in the media coverage of three main
earthquakes in 2015. The height of the boxes is proportional to the number of items (for each
earthquake). Box starts and ends corresponds to the first and third quartiles. The white line
inside corresponds to the median.