# Peer review of "Seismic Risk: The Biases of Earthquake Media Coverage"

_Geoscience Communication, 2019_

## Referee Comment (RC1) · Iain Stewart (Referee) · 9 May 2019

I really enjoyed reading this paper, which is a valuable analysis of the media responses to earthquake events and a considered appraisal of the media framing and key messaging that accompanies such seismic crises. It is fairly well written and concise, brings a strong interdisciplinary team to address the problem and sets the context well with a range of interesting background literature. The data collected is soundly analysed and well presented (I especially like Figure 4, a diagram which will probably be much used by risk communicators). To be honest, the paper isbroadly fit for publication as is, but I would suggest that the authors might like to make revisions around the following considerations:

[Figure]

Point 1: The thrust of the initial set up, not surprisingly, is the expectations of the media in disaster events. But the corollary is the epxectations of the role and responsibility of seismologists and scientists in those crisis moments. In this regard, I am thinking of Michelle Wood's work on actionable risk messaging. In regard, I wondered how much of the media responses analysised by the team incorporated expert comment and did that substantially change the messaging. This is important because it challenges the value and urgency of scientific expert comment during disasters, an aspect which the paper seems to omit. It may be beyond the scope of this studyy, but thoughts on this from the authors would be welcome.

Point 2: Your identification of an exponential decay of media interest seems to me an obvious but important point. It made me wonder if you could tie it to the predictable exponential decay in aftershock activity. I don't mean to suggest they are the same, or related, but conceptually or metaphorically it suggests the waning energy of the earthquake disaster. Just something to consider.

Point 3: One issue that does not seem to emerge from the media narratives documented in this study is 'where next?'. If true (and I suspect it is), this seems to me to be an important omission because coulomb stress triggering theory highlights the likelihood of transient dynamic stress being transferred to neighbouring faults and therefore increasing the probability (in the short term) of a triggered quake nearby. Although not without risks in terms of public panic, conveying the dynamic nature of earthquakes as evolving threat events would seem to be a media narrative that earthquake scientists could develop with the news media.

Point 4: I think it would help to clearly state why an appreciation of 'earthquake intensity' is better than an appreciation of 'earthquake magnitude'. Beyond the academic distinction, what is the utility for the public in those moments of crisis? Are we just being pedants about terminology, or is there a tangible public benefit in being explicit about using terms conveying energy and shaking?

Point 5: I'd love it if the paper could conclude with some recommendations to scientists about the key actionable risk messages that they ought to be conveying the media in the various time windows as an earthquake disaster unfolds, i.e. minutes-hours; hours-days; days-weeks (perhaps tied into a modified reprise of Figure 4. Recognising the likely changing media environment, how can scientists take more control over the narrative, particularly in the aftermath of the search and rescue operations where interest is dying down but seismic risk is potentially still high on neighbouring seismic sources?

Finally, some very minor points:

You refer to 'the media' but essentially it is the 'news media' and possibly even just the 'broadcast news media" that you are considering (e.g. not long-form documentaries etc.)

I'm not sure I know what you mean by 'the concept of the seismic crisis'

Figure 5 – the caption ought to explain the percentages. Some readers will no doubt be expecting the columns not add up to 100% and will be confused.

None of these points are especially substantive - they probably reflect my personal perspectives on this topic - and should not hamper publication of the very nice paper. My congratulations to the authors.

---

## Referee Comment (RC2) · Lisa Matthias (Referee) · 14 May 2019

Dear Editors,

Herein is my response to the manuscript entitled "Seismic Risk: The Biases of Earthquake Media Coverage", by Maud H. Devès and colleagues to Geoscience Communication. The authors present an interesting piece about international news outlets' reporting on earthquakes. I really like the figures as they nicely complement the paper – especially Figure 4, which visualizes and summarizes parts of the findings. However, I would suggest the authors describe in more detail the theoretical background of their paper, as well as their conceptual framework, and methodology. My relevant expertise for reviewing this paper is in framing research.

[Figure]

Lines 72 – 77: The first paragraph could do with some more clarity and explanations: a) I would use "media coverage" when first mentioning the term, variations are fine afterward; b) Public opinion about what? Some references would be great too here to support your arguments; c) Social media and online press are two very distinct things. I am not sure why the authors are mentioning "social media" at all since the news outlets they examine later on are traditional online news outlets, not social media platforms like Facebook, Twitter, or Reddit; d) "One would expect", who would expect, why would they expect this? The authors could draw on literature about the media's role in society, for example. See, for example: Lasswell (1948), Wright (1960).

Lines 79 – 80: "Scientists often blame journalists," making such a strong and generalized claim, I would add at least (!) two more references. And perhaps consider a less aggressive turn of phrase for balance.

Line 86: Please add a short explanation what is meant by "media filter."

Lines 79-95: I like that this paragraph zooms in on geoscience research in the media, but it would be good to contextualize these findings within the wider media sphere and its practices. Since the authors focus on "international" media, it would be enough to focus on general characteristics of media, such as news values (while cautioning that taxonomies of news values cannot explain everything). See, for example: Harcup and O'Neill (2001), Harcup and O'Neill (2017), Wu (2000).

When citing Harris' research, I am missing the explicit connection to framing research. For example, simplifying complex arguments is one of the very goals of framing, and so is the suggestion of a particular interpretation of events.

Moreover, in this context, it is important to note that, depending on the country, science journalism is declining (Bauer et al., 2013), and that non-specialist reporters are now covering science-related news, and that this of course contributes to the kind of coverage (e.g., how detailed the report will explain the research) the reader gets.
Line 93: Uncertainty about what?

Lines 97 – 107: I think this paragraph would actually fit much better right after the first (after line 77), if "Things, however, have proven to be more complex." was deleted, the text would also flow much more nicely. How the authors structure their text is of course up to them, but as a reader I was hoping for an explanation and/or some examples regarding the media's influence on public opinion and action in this context. I would find it easier to follow the manuscript if related paragraphs would be grouped together, and my mind would not have to jump between topics and then back.

Line 105 – 106: Direct implications for what? What agencies?

Lines 109 – 125: The relevance of this paragraph for the current paper should be highlighted.

Lines 127 – 161: Personally, I find this section difficult to follow. It is titled "This study," yet only 17 lines are about this study. I am missing explicit research questions and hypotheses, something that explains how the authors are approaching their overarching research question "in a globalized world, can we find systematic trends in how the international press covers earthquake events?"

Lines 147 – 148: a) The authors write "the different laws postulated by Galtung," but then cite Koopmans & Vliegenthart, 2010: Why not cite the original article? b) However, I am unsure why this is mentioned as the authors do not analyze, nor connect their findings to, the level of newsworthiness according to Galtung's taxonomy c) If I have missed the connection between Galtung and the authors' findings, there have been, at least, two notable "updates" on Galtung's work, which might be worth considering, Harcup and O'Neill (2001) and Harcup and O'Neill (2017).

Line 168: Could the authors please define "the international press" in the context of their paper, and also give a more detailed explanation for the selection of news outlets? In particular, I would argue that some news sources chosen for this paper do

not necessarily constitute as international news outlets, depending on how the term is defined: Vancouver Sun (looking at the circulation, it even becomes difficult to say this is a national newspaper), The Star, LA Times, El Informador. Likewise, it would be helpful to explain why these news sources were chosen over others that are arguably more relevant "international" news outlets (e.g., CNN, the BBC, Al Jazeera). Moreover, I would be interested to know why the authors included the Financial Times. While this is certainly an international news source, its focus is on business and economics.

Lines 168 – 169: Please briefly explain what data the geophysical set contains and how it was selected.

Line 169: The acronym "USGS" should be spelled out when used the first time.

Line 173: How did the authors assess "media quality?" What were the inclusion and exclusion criteria for RSS feed regularity, the geographical location of the news outlet, and volume? Why did the authors not include website traffic/news circulation in their source selection criteria?

Line 175: It would be good if the authors could elaborate on "sufficiently homogeneous:" What are the similarities, and where do the selected sources differ?

Line 183: Please add (n=X) for items that were excluded because they were void of relevant information? The authors should also state how many items they started out with, before the cleaning and tagging.

Lines 187 – 188: Please add how many duplicates were removed.

Line 190: What software did the authors use for the tagging process?

Line 193: Is the dictionary available somewhere?

Line 230-232: Critical discourse analysis does not just analyze texts but relates the content and its meaning to underlying structures of the sociopolitical context. This is also being done in Cox et al. (2008), which the authors say "inspired" their methodol-
ogy. For the context of this paper, it might be better to not use the term as this is not what is done.

Line 235: "As we are dealing with hundreds of thousands of items": In line 198, the authors write "4411," so this seems like a slight exaggeration. (see comment to line 544)

Lines 244 – 256: I like that the authors briefly and clearly describe the individual categories.

Line 270: Please briefly explain those limitations.

Line 390 – 391: Since the authors write "centers," which other regional centers have been referred to?

Line 419: What do the authors mean by "romanticized?"

Lines 476 – 477: This is really interesting, and it would be great if the authors could add a few quotes.

Line 484: Could the authors either explain further, or delete, "that one could call topoi?"

Lines 490 – 492: This is really interesting! Is this the same across all news outlets (i.e., do all, or a great proportion of, news outlets cover these topics for the Nepal earthquake?)? Why do the authors think this is?

Lines 521 – 522: Do the authors think that this might have been different if they had looked at news outlets from the countries the earthquakes were located in?

Line 544 (Figure 5): This might relate to my confusion in line 235, what does "items" refer to here? It seems that the authors are using it for different purposes (i.e., news items and ?)?!

Lines 553 – 554: Do the authors have any idea why? Since these are foreign news outlets, referring to celebrities could increase the newsworthiness of the reports?!
Lines 574 – 576: Again, I wonder if this might be different with local news outlets (i.e., the country affected by the earthquake) because "issues of recovery, restoration, reconstruction, adaptation, mitigation and preparedness" might seem somewhat more relevant to those countries than to faraway places, especially those that do not experience earthquakes.

References

Bauer, Martin W., Howard, Susan, Romo Ramos, Yulye Jessica, Massarani, Luisa and Amorim, Luis (2013) Global science journalism report: working conditions & practices, professional ethos and future expectations. Our learning series. Science and Development Network, London, UK.

Harcup, T., & O'neill, D. (2001). What Is News? Galtung and Ruge revisited. Journalism Studies, 2(2), 261-280.

Harcup, T., & O'neill, D. (2017). What is news? News values revisited (again). Journalism Studies, 18(12), 1470-1488.

Lasswell, H. D. (1948). The structure and function of communication in society. The communication of ideas, 37, 215-228.

Wright, C. (1960). Functional Analysis and Mass Communication. The Public Opinion Quarterly, 24(4), 605-620.

Wu, H. D. (2000). Systemic determinants of international news coverage: A comparison of 38 countries. Journal of communication, 50(2), 110-130.

---

## Author Comment (AC1) · 4 Jul 2019

Dear Ian Stewart,

First of all, we would like to thank you for your rigorous, thoughtful and constructive review. It was a pleasure to work at a response.

We prepared a point by point answer below. You can consult the details of the changes in the document entitled Devesetal_GC2019_revised_withtrackchanges. The final version of the revised manuscript, entitled Devesetal_GC2019_revised, has been compiled by accepting all changes. It will be uploaded on the website.

We believe that the suggested changes have significantly improved the paper and we hope you will find it even more ready than before for publication. We remain at your

disposal for any further improvements you might find necessary.

Sincerely,

Maud Devès on the behalf of all co-authors

— Point by point response

IS: "I really enjoyed reading this paper, which is a valuable analysis of the media responses to earthquake events and a considered appraisal of the media framing and key messaging that accompanies such seismic crises. It is fairly well written and concise, brings a strong interdisciplinary team to address the problem and sets the context well with a range of interesting background literature. The data collected is soundly analysed and well presented (I especially like Figure 4, a diagram which will probably be much used by risk communicators). To be honest, the paper is broadly fit for publication as is, but I would suggest that the authors might like to make revisions around the following considerations: Point 1: The thrust of the initial set up, not surprisingly, is the expectations of the media in disaster events. But the corollary is the expectations of the role and responsibility of seismologists and scientists in those crisis moments. In this regard, I am thinking of Michelle Wood's work on actionable risk messaging. In regard, I wondered how much of the media responses analysed by the team incorporated expert comment and did that substantially change the messaging. This is important because it challenges the value and urgency of scientific expert comment during disasters, an aspect which the paper seems to omit. It may be beyond the scope of this study, but thoughts on this from the authors would be welcome."

Authors answer: What should be the role and responsibilities of scientists in the face of disasters is a fundamental question. In this paper, we settle for exploring the media coverage of seismic events with the idea that it might help scientists to, at least, communicate more efficiently. News do not actually refer to scientists, specialists or to scientific explanation as much as what we expected before to undertake the study: only 5.4% of the news refer to the category we called 'experts', (Figure 5, table 2). And

the content of these references is, scientifically speaking, quite disappointing. Most of the time, it is just about mentioning the magnitude, in the best case, mentioning that earthquakes occur at plate boundaries. It is also very important to realize that these messages are mainly found in the initial phase of coverage i.e. immediately after main shocks or big aftershocks when journalists lack information to really build a story. As soon as more information comes about the level of impact, the first political declarations, etc., scientific considerations disappear. But a temporary lack of information is a void to be filled up... why not considering filling it up with a bit of scientific culture? The fact that there is only a very short time window (few hours in most cases) to communicate is an important result to that respect. We observed that the most cited expert institution was the USGS. One of us has recently had the chance to visit the news room of the French newspaper Le Figaro. He observed that, regarding earthquakes, journalists were using the information forwarded by press agencies (AFP-Reuters), the latter publishing automatically the communicates emitted by the USGS. Why not making, not just the USGS, but all scientific centers communicates more consistent from time to time?

IS: "Point 2: Your identification of an exponential decay of media interest seems to me an obvious but important point. It made me wonder if you could tie it to the predictable exponential decay in aftershock activity. I don't mean to suggest they are the same, or related, but conceptually or metaphorically it suggests the waning energy of the earthquake disaster. Just something to consider."

Authors answer: The referee makes a very interesting point. It is true that aftershocks big enough to be covered in the medias are less likely to occur as time passes by. We would be surprised however to find more in this statistical correlation. We tend to believe that exploring it is out of the scope of the current paper.

IS: "Point 3: One issue that does not seem to emerge from the media narratives documented in this study is 'where next?'. If true (and I suspect it is), this seems to me to be an important omission because coulomb stress triggering theory highlights the likelihood of transient dynamic stress being transferred to neighbouring faults and therefore increasing the probability (in the short term) of a triggered quake nearby. Although not without risks in terms of public panic, conveying the dynamic nature of earthquakes as evolving threat events would seem to be a media narrative that earthquake scientists could develop with the news media."

Authors answer: This is a very good point  We added a paragraph in the discussion as follows: "Another topic that is absent of media narratives is that of the location of the next event. Coulomb stress triggering theory can help answering that question, at least probabilistically speaking. It could thus be interesting to communicate on the dynamics of the seismic phenomenon, notably to help designing adequate prevention measures (it might shake elsewhere the next time!). Âż (lines 619-624)

IS: "Point 4: I think it would help to clearly state why an appreciation of 'earthquake intensity' is better than an appreciation of 'earthquake magnitude'. Beyond the academic distinction, what is the utility for the public in those moments of crisis? Are we just being pedants about terminology, or is there a tangible public benefit in being explicit about using terms conveying energy and shaking?"

Authors answer: We completed the paragraph accordingly: "As discussed in a previous paper (Le Texier et al., 2016), the term of magnitude is commonly used as a synonym of intensity by the media. But the notion of intensity is the one that allows introducing the idea of differential damages paving the way to discuss mitigation and preparedness (earthquake-resistant construction, site effects, etc.)."

IS: "Point 5: I'd love it if the paper could conclude with some recommendations to scientists about the key actionable risk messages that they ought to be conveying the media in the various time windows as an earthquake disaster unfolds, i.e. minutes-hours; hours-days; days-weeks (perhaps tied into a modified reprise of Figure 4. Recognising the likely changing media environment, how can scientists take more control over the narrative, particularly In the aftermath of the search and rescue operations where

interest is dying down but seismic risk is potentially still high on neighbouring seismic sources?"

Authors answer: This is a difficult question. Our analysis can provide a solid ground to design better scientific communication (notably by emphasizing the constraints linked to the large-scale dynamics of media coverage), but we do not have "the good recipe" to build the content of this communication and we tend to believe that this is an issue that falls out of the scope of our paper.

IS: "Finally, some very minor points: You refer to 'the media' but essentially it is the 'news media' and possibly even just the 'broadcast news media" that you are considering (e.g. not long-form documentaries etc.) I'm not sure I know what you mean by 'the concept of the seismic crisis'."

Authors answer: We agree with the referee on these two points. We modified occurrences of 'the media' in 'the news media' whenever appropriate. About the point on seismic crisis: we observed that the 'news media' tend to treat each earthquake as an autonomous event (sometimes not referring to it as an aftershock). This might contribute to the representation of the seismic phenomenon as being a powerful, but unique, shock. We know that aftershocks are particularly dangerous, and it is important that exposed population understand that: 1) it might shake elsewhere the next time (issue of the next location) and 2) that it might be shaking again after the main shock (issue of the temporal distribution).

IS: "Figure 5 – the caption ought to explain the percentages. Some readers will no doubt be expecting the columns not add up to 100% and will be confused. None of these points are especially substantive - they probably reflect my personal perspectives on this topic – and should not hamper publication of the very nice paper."

Authors answer: We thank again the referee for his very useful comments. We modified the caption accordingly. It now reads: "Percentage of news mentioning a theme or topic. NB: One news item can include several themes and topics."

Please also note the supplement to this comment:
https://www.geosci-commun-discuss.net/gc-2019-5/gc-2019-5-AC1-supplement.pdf
* * *

---

## Author Comment (AC2) · 4 Jul 2019

Dear Lisa Matthias,

First of all, we would like to thank you for your rigorous and thoughtful and constructive review. We answer point by point below. You can consult the details of the changes in the document entitled Devesetal_GC2019_revised_withtrackchanges. The final version of the revised manuscript, titled Devesetal_GC2019_revised, has been compiled by accepting all changes. It will be uploaded on the website.

We believe that the suggested changes have significantly improved the paper and we hope you will find it even more ready for publication, but we remain at your disposal for any further improvements you might find necessary. Thanks again for your help in

dealing with this manuscript.

Sincerely,

Maud Devès on the behalf of all co-authors

— point by point response —

LM: "Dear Editors, Herein is my response to the manuscript entitled "Seismic Risk: The Biases of Earthquake Media Coverage", by Maud H. Devès and colleagues to Geoscience Communication. The authors present an interesting piece about international news outlets' reporting on earthquakes. I really like the figures as they nicely complement the paper – especially Figure 4, which visualizes and summarizes parts of the findings. However, I would suggest the authors describe in more detail the theoretical background of their paper, as well as their conceptual framework, and methodology. My relevant expertise for reviewing this paper is in framing research. Lines 72 – 77: The first paragraph could do with some more clarity and explanations: a) I would use "media coverage" when first mentioning the term, variations are fine afterward; b) Public opinion about what? Some references would be great too here to support your arguments; c) Social media and online press are two very distinct things. I am not sure why the authors are mentioning "social media" at all since the news outlets they examine later on are traditional online news outlets, not social media platforms like Facebook, Twitter, or Reddit; d) "One would expect", who would expect, why would they expect this? The authors could draw on literature about the media's role in society, for example. See, for example: Lasswell (1948), Wright (1960)."

Authors answer: now lines 72-79. a) We agreed and modified the text accordingly. b) c) and d) We modified the text to include the referee' comments. Many substantial works have been published on the issue of the media's role in society. We added a reference to a recent work by Harcup and O'Neill (2017). For the rest, we prefer mentioning studies more directly related to disaster risk reduction. We added two more references: one about the role of the media in influencing everyone representations

about disasters; and another one, more operational, about the role of media in disaster risk reduction according to risk managers (Cottle, 2014 and Thanthathep et al., 2016). We still think it is important to mention social media and we hope the sentence placed at the very beginning of the paragraph will help clarify the focus of the paper from the outset. Newspapers are more and more influenced by the fact that the news they publish are further disseminated or shared on Twitter or Facebook but they remain the major gatekeepers in the process of news selection and dissemination (Harcup & O'Neill, 2017).

LM: "Lines 79 – 80: "Scientists often blame journalists," making such a strong and generalized claim, I would add at least (!) two more references. And perhaps consider a less aggressive turn of phrase for balance. Line 86: Please add a short explanation what is meant by "media filter." Lines 79-95: I like that this paragraph zooms in on geoscience research in the media, but it would be good to contextualize these findings within the wider media sphere and its practices. Since the authors focus on "international" media, it would be enough to focus on general characteristics of media, such as news values (while cautioning that taxonomies of news values cannot explain everything). See, for example: Harcup and O'Neill (2001), Harcup and O'Neill (2017), Wu (2000). When citing Harris' research, I am missing the explicit connection to framing research. For example, simplifying complex arguments is one of the very goals of framing, and so is the suggestion of a particular interpretation of events. Moreover, in this context, it is important to note that, depending on the country, science journalism is declining (Bauer et al., 2013), and that non-specialist reporters are now covering science-related news, and that this of course contributes to the kind of coverage (e.g., how detailed the report will explain the research) the reader gets. Line 93: Uncertainty about what?"

Authors answer: Lines 93-108. We reworked the paragraph in order to take into account the referee' comments.

We rephrased the first sentence and added one more reference to a paper about the l'Aquila trial (Cocco et al., 2015), an event that transformed deeply the relationship of the geoscience community to the press. We also added an explicit mention to the other papers discussed in the paragraph (we forgot to list them in the first sentence in the earlier version of the paper). Our idea in this paragraph is to show that media analysis is an important issue for geoscientists.

About the concept of 'media filter', we rephrased the sentence in order to make it clearer.

The referee is right when she says that the geoscientists we cite do not pay enough attention to the media rules and habits. But we did not want to add that discussion at the beginning of the paper as it might seem to technical to non-specialists. Again our idea here was really to show that media analysis (i.e. also understanding how the media work) is an important issue for geoscientists. We cut bits and pieces about Harris' work. Hopefully, it will make the argument clearer. The criteria of news selection formalized by Galtung & Ruge (1965) and further completed by Harcup & O'Neill (2001, 2017) indicates a clear preference of journalists for "bad news", "conflicts", "surprise" or "drama" and a clear rejection of "complex" stories, especially in the case of daily newspapers. In the case of online newspapers, the need to get clicks and shares has undoubtedly reinforced the influence of these factors in decisions about what news to select, as well as news treatment (Harcup & O'Neill, 2017). This might also explain why science journalism is declining (Bauer et al., 2013).

We meant scientific uncertainties and modified the sentence to make it more explicit.

LM: "Lines 97 – 107: I think this paragraph would actually fit much better right after the first (after line 77), if "Things, however, have proven to be more complex." was deleted, the text would also flow much more nicely. How the authors structure their text is of course up to them, but as a reader I was hoping for an explanation and/or some examples regarding the media's influence on public opinion and action in this context. I would find it easier to follow the manuscript if related paragraphs would be grouped together, and my mind would not have to jump between topics and then back. Line 105 – 106: Direct implications for what? What agencies?"

Authors answer: Lines 81-91. We agree with the referee's suggestion. We moved the paragraph and reworked the text accordingly, deleting notably the unhappy sentence: "Things, however, have proven to be more complex".

About agencies, we meant any agencies involved in disaster risk reduction. We modified the sentence as follows: "how [the] agencies [involved in disaster risk reduction] could reduce fatalism and facilitate preventive action by the way they present information about earthquakes and other disasters."

LM: "Lines 109 – 125: The relevance of this paragraph for the current paper should be highlighted."

Authors answer: We removed the paragraph.

LM: "Lines 127 – 161: Personally, I find this section difficult to follow. It is titled "This study," yet only 17 lines are about this study. I am missing explicit research questions and hypotheses, something that explains how the authors are approaching their overarching research question "in a globalized world, can we find systematic trends in how the international press covers earthquake events?""

Authors answer: Lines 110-179. We have modified the title of the section 1.2. to be more consistent with its content and attempted to formulate more clearly our research question. In a previous paper (Le Texier et al. 2016), we showed that the coverage of earthquakes in international news published by daily newspapers concentrated on a limited number of events due to differences in the geophysical, geographical and political contexts of the different earthquakes and demonstrated a strong homogeneity in the editorial selection process. This study really aims to look into the temporal dynamics of the coverage (duration, trends) and into the potential existence of a typical framing of 'earthquake news' (i.e. by comparing news content between events and between newspapers from various countries and languages).

LM: "Lines 147 – 148: a) The authors write "the different laws postulated by Galtung," but then cite Koopmans & Vliegenthart, 2010: Why not cite the original article? b) However, I am unsure why this is mentioned as the authors do not analyze, nor connect their findings to, the level of newsworthiness according to Galtung's taxonomy c) If I have missed the connection between Galtung and the authors' findings, there have been, at least, two notable "updates" on Galtung's work, which might be worth considering, Harcup and O'Neill (2001) and Harcup and O'Neill (2017)."

Authors answer: Lines 160-163. We modified the sentences as follow: "It is thus possible to analyze the level of newsworthiness according to the general laws postulated by Galtung and its followers (Galtung & Ruge, 1965; Harcup & O'Neill, 2001, 2017; Wu, 2000) and their specific application to earthquake media coverage (Koopmans & Vliegenthart, 2010)."

LM: "Line 168: Could the authors please define "the international press" in the context of their paper, and also give a more detailed explanation for the selection of news outlets? In particular, I would argue that some news sources chosen for this paper do not necessarily constitute as international news outlets, depending on how the term is defined: Vancouver Sun (looking at the circulation, it even becomes difficult to say this is a national newspaper), The Star, LA Times, El Informador. Likewise, it would be helpful to explain why these news sources were chosen over others that are arguably more relevant "international" news outlets (e.g., CNN, the BBC, Al Jazeera). Moreover, I would be interested to know why the authors included the Financial Times. While this is certainly an international news source, its focus is on business and economics. Line 173: How did the authors assess "media quality?" What were the inclusion and exclusion criteria for RSS feed regularity, the geographical location of the news outlet, and volume? Why did the authors not include website traffic/news circulation in their source selection criteria? Line 175: It would be good if the authors could elaborate on "sufficiently homogeneous:" What are the similarities, and where do the selected sources differ?"

Authors answer: Lines 184-198. We added an explicit definition of what we call "international news" at the very beginning of section 1.2 (lines 143-145). It states: "The current paper focusses on 'international news'. By 'international news', we mean news published by daily newspapers about foreign countries or, in practical terms, news published by newspapers through a specific RSS flows entitled "international" or "world"." We have systematically replaced "international press" by "international news published by daily newspapers" or, in abbreviate form, by "international news".

The details of the criteria used to build the media corpus are discussed in the supplementary of Grasland (2019). We added the reference that we forgot in the earlier version in the presentation of the datasets. It can be consulted here: https://journals.sagepub.com/doi/suppl/10.1177/1748048518825091/suppl_file/Supplemental_Material.pdf.

The term of "media quality" has been replaced by "national or international status of newspapers (broadsheet newspapers)". We removed "sufficiently homogeneous" from the sentence. The newspapers entering the database are newspapers with important audience in their home country that play an important role in the importation of foreign news from the rest of the world. Some broadsheet newspapers have also been selected for their global audience (e.g. Financial Times).

LM: "Lines 168 – 169: Please briefly explain what data the geophysical set contains and how it was selected."

Authors answer: Lines 184-198. For the geophysical database, we use the seismic catalogue provided by the USGS. The USGS collects and analyses data recorded by several networks of seismographs throughout the world. It maintains an online catalogue of archives called ANSS (Advanced National Seismic System) Comprehensive Catalog. This is, to date, the most exhaustive database freely accessible to the general public. The catalog is accessible here: https://earthquake.usgs.gov/earthquakes/. It is well-known by geoscientists and we were not sure it would be necessary to develop much further in the main text but we added a sentence for the sake of clarity: "The geophysical dataset is built from the online seismic catalogue provided by the United States Geological Survey (ANSS). For each earthquake, we collect the following parameters: hypocenter, magnitude and label."

The ANSS catalogue offers access to the main parameters that allow geophysicists to characterize earthquakes. Among these, the parameters we have retained for comparison with the media database are the following: • The hypocenter localizes the seismic source in terms of latitude, longitude, and depth. • The magnitude measures the energy liberated by the earthquake. In other terms, it is the measure of its objective "physical importance". However, there are different ways to estimate the magnitude of an event and different types of magnitude can follow each other over time. The USGS catalog prefers the moment magnitude (Mw). For recent periods, such as the one used in this study, the magnitude indicated in the database is therefore in theory a moment magnitude. Should the USGS send press releases immediately after receiving the preliminary data for seismic events of magnitudes over 6, the event's magnitude is generally re-evaluated with the arrival of additional data. Therefore, the press mentions different magnitudes for the same event. • The label offers the geographic localization of the event with two different parameters. One is a political (country, region) or geographic (continent, sea, ocean) variable. The other is a spatial variable of distance and orientation in terms of the nearest population center with over 1,000 inhabitants within a radius of 300 kilometers. Though they may seem redundant with the precise geographic coordinates of latitude and longitude, these variables are extremely important in terms of the media since they offer the earthquake not only a position but also a nationality and a location allowing the public to name and memorize it. We will see that there is confusion as several earthquakes (e.g. main shock and aftershocks) are defined as different events by Earth scientists but are often conflated from a political and media standpoint.

LM: "Line 169: The acronym "USGS" should be spelled out when used the first time."

Authors answer: It has been done. See line 187.

LM: "Line 183: Please add (n=X) for items that were excluded because they were void of relevant information? The authors should also state how many items they started out with, before the cleaning and tagging. Lines 187 – 188: Please add how many duplicates were removed."

Authors answer: All numbers are indicated in Figure 2. We do not find necessary to repeat the information but we can if the referee finds it necessary.

LM: "Line 190: What software did the authors use for the tagging process?"

Authors answer: The software R was used for all the analyses. We used notably the package tm for text analysis. We added a sentence at the end of section 2.1 (lines 197-198).

LM: "Line 193: Is the dictionary available somewhere?"

Authors answer: We tagged news related to earthquakes using the following dictionary. Error rates are given in the paper lines (217-218). They are reasonably small (4% for false positives, 2 to 3% for false negatives).

word type TAG language aftershock disaster_name Earthquake en aftershocks disaster_name Earthquake en temblor disaster_name Earthquake en temblors disaster_name Earthquake en seismic disaster_name Earthquake en seismicity disaster_name Earthquake en seism disaster_name Earthquake en seisms disaster_name Earthquake en tremor disaster_name Earthquake en tremors disaster_name Earthquake en tsunami disaster_name Earthquake en tsunamis disaster_name Earthquake en quake disaster_name Earthquake en quakes disaster_name Earthquake en earthquake disaster_name Earthquake en earthquakes disaster_name Earthquake en terremoto disaster_name Earthquake es terremotos disaster_name Earthquake es temblor de tierra disaster_name Earthquake es temblores de tierra disaster_name Earthquake es sismo disaster_name Earthquake es sismos disaster_name Earthquake es seísmo disaster_name Earthquake es seísmos disaster_name Earthquake es sismico disaster_name Earthquake es sismica disaster_name Earthquake es sismicidad disaster_name Earthquake es tsunami disaster_name Earthquake es tsunamis disaster_name Earthquake es maremoto disaster_name Earthquake es maremotos disaster_name Earthquake es aftershock disaster_name Earthquake es aftershocks disaster_name Earthquake es temblor disaster_name Earthquake es temblores disaster_name Earthquake es tremblements de terre disaster_name Earthquake fr tremblement de terre disaster_name Earthquake fr séisme disaster_name Earthquake fr séismes disaster_name Earthquake fr sismique disaster_name Earthquake fr sismiques disaster_name Earthquake fr tsunami disaster_name Earthquake fr tsunamis disaster_name Earthquake fr aftershock disaster_name Earthquake fr aftershocks disaster_name Earthquake fr

We believe that as the response to referees will be available online, there is no need to add this table in the paper (it is a bit long and boring to read). But we will follow the reviewers' advices on the subject.

The country dictionary used in the paper is available on request at claude.grasland@parisgeo.cnrs.fr. It is specific to the year 2015 (names of state representatives, etc.) and limited to three languages (FR, EN, SP). A less precise but more polyvalent dictionary can also be found in the R package newsmap by Kohei Watanabe: https://github.com/koheiw/newsmap.

LM: "Line 230-232: Critical discourse analysis does not just analyze texts but relates the content and its meaning to underlying structures of the sociopolitical context. This is also being done in Cox et al. (2008), which the authors say "inspired" their methodology. For the context of this paper, it might be better to not use the term as this is not what is done."

Authors answer: Line 252. We removed the reference to "critical discourse analysis".

LM: "Line 235: "As we are dealing with hundreds of thousands of items": In line 198, the authors write "4411," so this seems like a slight exaggeration. (see comment to line 544)"

Authors answer: Line 254. By Âń hundreds of thousands items Âż we refered to the total EQmedia database. But we agree with the referee that this can introduce some misunderstanding as the detection of "textual silences" and key words has only been done on the 4441 items mentioning earthquakes. We modified the text accordingly.

LM: "Lines 244 – 256: I like that the authors briefly and clearly describe the individual categories."

Authors answer: Thanks 

LM: "Line 270: Please briefly explain those limitations."

Authors answer: Lines 292-293. We added explanations in section 2.3. There are limitations to the keyword approach: the meaning of isolated words is often ambiguous and related to the context and the position before or after other words (Church & Hanks, 1990). But the independent classification of the items by the coauthors indicates a good consistency in the coding of themes and subthemes and the identification of topics (we reach a maximum of 12% of differences for the emergency response category). More details on how to improve our "bags of words" approach are provided in the final discussion of the paper.

LM: "Line 390 – 391: Since the authors write "centers," which other regional centers have been referred to?"

Authors answer: Line 413. The sentence was unfortunate. We reformulated it as follows: national meteorological agencies and emergency operations centers, etc.

LM: "Line 419: What do the authors mean by "romanticized?""

Authors answer: We deleted Âń the event starts to be romanticized Âż as we were not sure of the translation from French to English. . . We meant: "being embedded into a story (often partly imaginary)".

LM: "Lines 476 – 477: This is really interesting, and it would be great if the authors could add a few quotes."

Authors answer: Lines 499-505. We added two quotes for illustration.

LM: "Line 484: Could the authors either explain further, or delete, "that one could call topoi?"" Authors answer: Agreed and done.

LM: "Lines 490 – 492: This is really interesting! Is this the same across all news outlets (i.e., do all, or a great proportion of, news outlets cover these topics for the Nepal earthquake?)? Why do the authors think this is?"

Authors answer: The most important difference in terms of news content seems to be linked to the duration of coverage. The coverage of the Nepal quake lasts longer, and news are richer.

LM: "Lines 521 – 522: Do the authors think that this might have been different if they had looked at news outlets from the countries the earthquakes were located in?"

Authors answer: We focused on international news and cannot answer directly to the question. But we know proximity does matter. This is suggested by the case of the Times of India which has the largest and longer coverage of the Nepal quake.

LM: "Line 544 (Figure 5): This might relate to my confusion in line 235, what does "items" refer to here? It seems that the authors are using it for different purposes (i.e., news items and ?)?!"

Authors answer: The referee is right. We use sometimes "news items", "items" or "news". All these versions actually mean the same. We have corrected the text and the figures accordingly.

LM: "Lines 553 – 554: Do the authors have any idea why? Since these are foreign news outlets, referring to celebrities could increase the newsworthiness of the reports?!"

Authors answer: Lines 580-582. We agree with the referee. It confirms classical rules of news value about elite people and celebrities formulated by Galtung and Rudge (1965) and Harcup and O'Neill (2001, 2017). We added a sentence spelling out that point.

LM: "Lines 574 – 576: Again, I wonder if this might be different with local news outlets (i.e., the country affected by the earthquake) because "issues of recovery, restoration, reconstruction, adaptation, mitigation and preparedness" might seem somewhat more relevant to those countries than to faraway places, especially those that do not experience earthquakes."

Authors answer: A very good point, but out of the scope of the current paper.

Please also note the supplement to this comment:
https://www.geosci-commun-discuss.net/gc-2019-5/gc-2019-5-AC2-supplement.pdf
* * *
[Figure]

**Supplement:**

[revised manuscript text omitted]

EN: magnitude, Richter
SP: grados, Richter, magnitud(es)
FR: magnitude, Richter

**Tsunami**
EN: tsunami(s)
SP: tsunami(s), maremoto(s), olas
FR: tsunami(s)

**Aftershocks**
EN: aftershock(s)
SP: aftershock(s), réplica(s)
FR: aftershock(s), réplique(s)

**Other secondary hazards**
EN: avalanche(s), landslide(s), flood(s)/flooding
SP: avalancha(s), deslizamiento(s), alud, inundacion(es)
FR: glissement(s) de terrain, avalanche(s) |
| **IMPACTS** | **Impacts – general**
EN: hit(s), struck, felt, shook, shak(e)(ing)(en), rocked, jolt(s)(ed), rattled, shattered, sway(ed), battered, suffered, toppling, crushed, strike, stricken, impact
SP: impacto, estimacion(es), afectación, sacud(e)(ido)(ida)(idas)(ieron), golp(e)(eó)(ea), golpead(o)(os)(a)(as), azotó, azotado, sentido, se sintió, afectó, sufrieron, arrasó, temblar, asoló, castigad(o)(a)
FR: frappé(e), touché(s), ressenti(e), ébranlé, secoué

**Human impact**
*Human impact – general*
EN: fatalities, casualt(y)(ies), victim(s), affected, stranded
SP: balance, víctima(s), afectados, damnificados, recuento(s), saldo, contabilizado
FR: bilan, victime(s), sinistrés
*Human impact – death toll*
EN: death(s), kill(s)(ed)(ing), dead, bodies, died, deadly, claimed
SP: muerto(s), muerte(s), mueren, murieron, mortal(es), fallecido(s), fallecieron, cuerpos, cadavers, decesos, mató
FR: mort(s), tué(e)(s), corps, meurtrier
*Human impact – injured*
EN: injured, wounded
SP: heridos
FR: blesses

**Material damage**
*Material damage – general*
EN: rubble, damage(d), collaps(e)(es)(ed) (ing), devastat(ed)(ion), destroy(ed)(ing), destruction, wreckage, debris, ravaged, ruins/ruined
SP: daños, escombros, dañad(os)(as), destruid(o)(os)(as), perdidas, destrucción, ruinas, caíd(o)(a), destruyó, destrozadas, colapso, devastó, devastadas, derrumb(e)(es)(aron)(ado)
FR: dévast(é)(ée), décombres, dégâts, détruit/détruits, effondr(ée)(ées), destructions, gravats |

| | | |
|---|---|---|
| | *Material damage - on buildings*
EN: homes, building(s), houses, structure(s), property
SP: edificio(s), vivienda(s), edificaciones
FR: maisons, bâtiments

*Material damage - on infrastructures*
EN, FR: no recurrent keywords were found
SP: eléctricas, infraestructuras | |
| **EMERGENCY RESPONSE** | **Tsunami warning**
EN: tsunami warning(s), alert(s)
SP: alerta de tsunami, alarma
FR: alerte

**Evacuation**
EN: evacuat(e)(ed)(ion)(ions), evacuees
SP: evacuad(os)(as), evacuar, evacuación
FR: evacu(ees)(er)(ation)

**Aid, Search & Rescue**
*General*
EN: effort(s), response, respond, operation(s), deployed, aid, rescu(e)(es)(ed)(ing), relief, help(ed)(ing), assist(ance), helicopter(s), chopper, aircraft, support, send(s)(ing), save(d), distribut(ing)(ion), airlifted, dig(ging), dug, missing, search(ing), alive, pulled, trapped, recovered + table 2/rescuers
SP: operación/operaciones, gestión, respuesta, solidaridad, crisis, apoy(o)(ar), ordenó, responder, envoi, enviado(s), reacción, ayuda, ayudar, ayudas, ayudando, rescate, rescatar, rescatan, rescatado, helicóptero(s), asistencia, socorro, attender, ofrece, aeronave, búsqued(a)(as) + table 2/rescuers
FR: operation(s), répondre, secours, aide, sauver, assistance, disparu, chiens, recherchés, sans nouvelles + table 2/rescuers
*Vital needs and supplies*
EN: food, hungry, sanitation, water, drink(ing), fuel, blankets, gasoline, suppl(y)(ies), resources, basic, vital, lack of, goods, need, needed, material, equipment
SP: agua, alimentos, alimentaria, necesidad(es), comida, suministro(s)
FR: de materiel, besoins

**Medical care**
EN: hospital(s), medical, medicine(s), disease(s), health, outbreak, epidemic(s), treatment, patients
SP: hospital(es), médico(s), salud, medicinas, sanitarios
FR: no recurrent keywords were found

**Displacement & Temporary shelter**
EN: shelter(s), outdoors, sleep, sleeping, homeless, refuge, fled
SP: noche al raso, albergues, tiendas de campaña, desplazados, refugio(s)
FR: camps, fuir, dehors

**Cremation**
EN, FR: no recurrent keywords were found
SP: funerarias | |
| **RECOVERY REHABILITATION RECONSTRUCTION** | **Recovery/Reconstruction**
EN: recover(y)(ing), return to, returned, reconstruction, rebuild(ing), reopen(s)(ed), normal | |

| | |
|---|---|
| **(PREPAREDNESS)** | SP: desescombro, reconstrucción, reconstruir, normalidad
FR: reconstruction

*No recurrent keywords were found that unambiguously refer to Risk assessment, development and land use planning / Adaptation and mitigation measures / Education and information / Preparedness, contingency planning, consolidate preparations for next disasters* |

**Table 2.** Identity matrix. Contains the keywords used to quantify the presence/absence of
different categories of stakeholders. Keywords were identified from a list of most frequent
words using different thresholds for English, Spanish and French to balance differences in the
RSS feed numbers.

| CONTENT CATEGORIES | KEYWORDS BY THEMES AND TOPICS |
|---|---|
| **STATES** | EN: nation, state(s), government(s), authorities, minister(s), ministry, foreign secretary, foreign office, president, parliament, royal rulers, embassy, European Union
SP: país, nación, gobierno, autoridades, ministerio, ministro, president(a)(e), exteriores, funcionarios, gabinete, ispr, fata, europea
FR: pays, gouvernement, affaires etrangeres, autorités, ministère, ministre, Quai d'orsay |
| **UN AGENCIES** | EN: United Nations, UNICEF, UNESCO, World Food Programme
SP: onu, naciones unidas, Programa Mundial de Alimentos, unesco, unicef
FR: nations unies, onu |
| **INTERNATIONAL AID** | EN: international aid, international agencies, aid agencies, humanitarian aid
SP: ayuda internacional, comunidad internacional, organización no gubernamental, ong, cruz roja
FR: aide internationale, croix rouge, humanitaire(s) |
| **CIVIL SECURITY & DEFENSE** | EN: police, army, military, marine(s), air force, soldiers, troops, firefighters, Gurkhas
SP: ejército, policía, militares, armada, marina, soldados, Oficina Nacional de Emergencia |
| **RESCUERS** | EN: rescuers, rescue team(s), aid workers, rescue workers, relief workers, volunteer(s), personnel
SP: equipo de rescate, equipos de rescate, servicios de emergencia, rescatistas, socorristas
FR: équipe, secouristes, sauveteurs |
| **AFFECTED PEOPLE** | **Directly affected ones**
EN: people, rescued, survivor(s), victims, those affected
SP: persona(s), víctima(s), los afectados, damnificados, desaparecid(o)(a)(os)(as), supervivientes, sobrevivient(e)(es), rescatad(o)(os)
FR: victimes, survivant(s), sinistrés, rescapes, personnes
**Locals** |

| | |
|---|---|
| | EN: residents, locals, villagers, sherpa(s), guides, Famous locals: Ang Tshering, Bajracharya
SP: población, habitantes, guías
FR: habitants, villageois, population
**Vulnerable ones**
EN: children, child, boy, girl(s), wo(man)(men), famil(y)(ies), teenag(e)(er), teen, bab(y)(ies)
SP: niños, famili(a)(as), muj(er)(eres), jóven, bebe, anciano
FR: familles, adolescent, enfants, orphelins |
| **'EXPERTS'** | EN: expert(s), US Geological Survey, specialists, scientists
SP: usgs, Centro Sismológico Nacional, especialistas, Servicio Hidrográfico y Oceanográfico de la Armada
FR: usgs, institute américain de géophysique |
| **PRIVATE COMPANIES** | EN: Google, Facebook, compan(y)(ies)
SP: google, Facebook
FR: no recurrent keywords were found |

[Figure]

**Figure 4.** Temporal distribution of the media coverage for the three well-covered earthquakes
of the year 2015. The color scale allows comparing the duration of the media coverage with the
expected duration of the different phases of disaster risk management models.

| | Themes | % of earthquake news | Number of items | Subthemes | % | Number of items | Topics | % | Number of items |
|---|---|---|---|---|---|---|---|---|---|
| **Discourse content** | Hazard | 45,8 | 2020 | Tsunami | 8,9 | 391 | | | |
| | | | | Aftershocks | 5,8 | 254 | | | |
| | | | | Secondary hazards | 7,8 | 343 | | | |
| | | | | Magnitude estimation | 23,5 | 1036 | | | |
| | Impacts | 76,7 | 3384 | General impact | 40,9 | 1802 | | | |
| | | | | Human impact | 49,6 | 2189 | General | 17,1 | 756 |
| | | | | | | | Death toll | 40,7 | 1797 |
| | | | | | | | Injured | 8,9 | 393 |
| | | | | Material damage | 30,8 | 1358 | General | 26,1 | 1150 |
| | | | | | | | Buildings | 13,3 | 585 |
| | Response | 45,3 | 1996 | | 4,3 | 191 | | | |
| | | | | Evacuation | 2,1 | 93 | | | |
| | | | | Aid Search Rescue | 34,0 | 1501 | General | 29,6 | 1306 |
| | | | | | | | vital needs | 4,4 | 196 |
| | | | | Medical care | 2,2 | 95 | | | |
| | | | | Temporary shelter | 2,7 | 117 | | | |
| | Reconstruction | 5,6 | 249 | | | | | | |
| **Identity Matrix** | States | 27,7 | 1220 | | | | | | |
| | Un agencies | 3,8 | 168 | | | | | | |
| | International Aid | 2,5 | 111 | | | | | | |
| | Civil Security Defence | 8,0 | 353 | | | | | | |
| | Rescuers | 7,7 | 341 | | | | | | |
| | Affected People | 44,2 | 1951 | Directly affected ones | 33,4 | 1475 | | | |
| | | | | Locals | 4,8 | 211 | | | |
| | | | | Vulnerables | 6,0 | 265 | | | |
| | Expert | 5,4 | 239 | | | | | | |
| | Private Companies | 1,6 | 72 | | | | | | |

**Corpus = 320 888 news items, including 4 411 news items about earthquake (1,37%)**

**Figure 5.** Percentage of news items mentioning a theme or topic. NB: One news item can include several themes and topics.

[Figure]

**Figure 6.** Temporal distribution of the DRM categories in the media coverage of three main earthquakes in 2015. The height of the boxes is proportional to the number of news items (for each earthquake). Box starts and ends corresponds to the first and third quartiles. The white line inside corresponds to the median.